# Purity Law for Neural Routing Problem Solvers with Enhanced Generalizability

**Wenzhao Liu**[1]    **Haoran Li**[1]    **Congying Han**[1*]    **Zicheng Zhang**[2]

**Anqi Li**[3]    **Tiande Guo**[1]

[1]School of Mathematical Sciences, University of Chinese Academy of Sciences
[2]JD.com
[3]School of Mathematical Sciences, Nankai University

{liuwenzhao22,lihaoran21}@mails.ucas.ac.cn, hancy@ucas.ac.cn
zhangzicheng6@jd.com, anqili@nankai.edu.cn, tdguo@ucas.ac.cn

## Abstract

Achieving generalization in neural approaches across different scales and distributions remains a significant challenge for routing problems. A key obstacle is that neural networks often fail to learn robust principles for identifying universal patterns and deriving optimal solutions from diverse instances. In this paper, we first uncover Purity Law, a fundamental structural principle for optimal solutions of routing problems, defining that edge prevalence grows exponentially with the sparsity of surrounding vertices. Statistically and theoretically validated across diverse instances, Purity Law reveals a consistent bias toward local sparsity in global optima. Building on this insight, we propose Purity Policy Optimization (PUPO), a novel training paradigm that explicitly aligns characteristics of neural solutions with Purity Law during the solution construction process to enhance generalization. Extensive experiments demonstrate that PUPO can be seamlessly integrated with popular neural solvers, significantly enhancing their generalization performance without incurring additional computational overhead during inference. The code is available at `https://github.com/Kejun0627/PUPO`.

## 1 Introduction

Routing problems, such as Traveling Salesman Problem (TSP) and Vehicle Routing Problem (VRP), are fundamental combinatorial optimization (CO) problems with broad applications in operations research and computer science fields, including scheduling [38, 3], circuit compilation [36, 9], and computational biology [25, 15]. Due to their NP-hard nature, even the most advanced exact solver [2] could not efficiently find the optimal solution for large-scale instances within a reasonable time frame. As a result, approximate heuristic algorithms, such as LKH3 [16, 17] and HGS [42], have been developed to find near-optima with improved efficiency. However, these approaches still face significant computational overhead owing to their iterative search processes for each instance.

Deep learning, particularly deep reinforcement learning, holds significant promise for developing fast and advanced neural routing problem heuristics. Learning-based approaches generally fall into two categories: methods that learn to construct solutions step by step [4, 24, 26, 18], and learn to search for better solutions iteratively [30, 7, 11]. Among these, neural constructive methods excel in faster inference and higher performance, enabling real-time applications. However, they often struggle with generalization and exhibit limited performance on large-scale or heterogeneous instances.

---

*Corresponding author: hancy@ucas.ac.cn

39th Conference on Neural Information Processing Systems (NeurIPS 2025).

The primary challenge behind this weak generalizability lies in the tendency of neural models to overfit to specific patterns tied to particular training settings. To address this, some approaches have focused on simplifying the decision space, such as restricting the feasible action set [10, 12] and employing divide-and-conquer strategies [22, 37, 31]. While these methods improve the performance, they inherently reduce the potential optimality of the solutions. More importantly, they lack guidance from the universal structural properties of routing problems, which prevents neural solvers from directly learning consistent and applicable patterns across various instances during training.

In this paper, we first explore generalizable structural patterns in routing problems, then leverage them to guide the learning process of neural solvers. To begin with, we define edge purity order as a measure of vertex density around edges, where sparser neighborhoods yield lower purity orders. We reveal Purity Law, a fundamental principle for routing problems stating: *the proportion of different edges in the optimal solution follows a negative exponential law based on their purity orders across various instances.* This indicates that edges with lower purity orders are more prevalent in optimal solutions, with their frequency increasing exponentially as the order decreases. Purity Law demonstrates its potential to capture universal structural information in two key ways. First, it is ubiquitously present in optimal solutions in various instances, underscoring *its strong connection to optimality*. Second, the parameters of its negative exponential model remain statistically invariant across varying instance scales and distributions, proving that *the sparsity-driven edge dominance law is intrinsic rather than data-specific*.

Based on the universal presence of Purity Law, we propose **PU**rity **P**olicy **O**ptimization (PUPO), a novel training framework that incorporates this generalizable structural information into the policy optimization process. PUPO guides the solution construction process by encouraging the emergence of Purity Law. Specifically, it modifies the policy gradient to balance the purity metrics of solutions at different decision stages. By aligning with Purity Law, PUPO helps models learn consistent patterns independent of specific instances, thereby improving their ability to generalize to different distributions and scales, without altering the underlying network architecture. Notably, PUPO can be easily integrated with a wide range of existing neural solvers. Extensive experiments show that PUPO significantly enhances the generalization ability of these solvers, without increasing computational overhead during inference. Our contributions include:

- We identify Purity Law as a fundamental principle that reliably characterizes optima across various instances, offering a novel perspective to comprehend common structural patterns in routing problems.
- We propose PUPO, a novel training approach that explicitly encourages alignment with Purity Law during the solution construction process, helping models to learn consistent patterns across instances.
- We demonstrate through extensive experiments that PUPO can be seamlessly integrated with popular neural routing problem solvers to considerably improve their generalization, without increasing computational cost during inference.

## 2 Related Work

According to the way the solutions are generated, neural approaches can488 generally be divided into two classes: learning to directly construct and learning to iteratively improve solutions. Discussions on classical neural solvers are provided in Appendix A. Traditional Neural methods often struggle with poor generalization [19]. Recently, some works have attempted to improve the generalizability of neural routing problem solvers. UTSP [35] applies unsupervised learning by leveraging permutation properties of TSP, but fails to scale to instances larger than 2000 vertices. HTSP [37] improves efficiency for large-scale TSP by employing hierarchical reinforcement learning. LEHD [31] and GLOP [45] continuously refine solutions through divide-and-conquer strategies via neural models. Following local reconstruction, SIL [32] proposes self-improved training, utilizing the pseudo-labels of previous partial solutions, which boosting the scalability of NCO methods. INViT [10] simplify the decision space by restricting actions to local neighborhoods, as well as enforcing the invariant nested view transformer, to improve generalization. ELG [12] ensembles a global policy and local policies, whose outputs are aggregated with a pre-fixed rule. The local policies also utilizing k-nearest neighbors to promote cross-size generalizability. More detailed discussions are provided in Appendix A. However, these approaches inherently trade off the optimality of solutions in favor of generalization. Despite progress, no research has explored universal structural principles of routing problems to guide neural learning, leaving the challenge of identifying consistent instance patterns unsolved.

# 3 Preliminary

## 3.1 Routing Problems

In this paper, we focus on the Euclidean routing problems in two-dimensional space. Given an instance, let $G = (\mathcal{X}, \mathcal{E})$ represents an undirected graph, $\mathcal{X} = \{x_i | 1 \leq i \leq |\mathcal{X}|\}$ is the vertex set, and $\mathcal{E} = \{e_{ij} = (x_i, x_j) | 1 \leq i, j \leq |\mathcal{X}|\}$ is the edge set, where $|\mathcal{X}|$ is the number of vertices. In the Euclidean routing problems, $G$ is fully connected and symmetric, with each vertex $x_i \in \mathcal{X}$ represented by a coordinate $(x_i^1, x_i^2)$ scaled to the unit square $[0, 1]$. Each edge $e_{ij} \in \mathcal{E}$ is assigned a visit cost $c(x_i, x_j)$, typically the Euclidean distance between the vertices $x_i$ and $x_j$. A feasible solution $\boldsymbol{\tau} = (\tau_1, \ldots, \tau_N)$ is an index sequence of length $N$ that satisfies all the constraints. The objective is to find the optimal one with the minimum total cost among all feasible solutions, where the total cost of $\boldsymbol{\tau}$ is formulated as follows:

$$L(\boldsymbol{\tau}) = c(x_{\tau_N}, x_{\tau_1}) + \sum_{i=2}^{N} c(x_{\tau_{i-1}}, x_{\tau_i}). \tag{1}$$

The constraints vary according to specific routing problems. For example, in TSP, the constraint is that feasible solutions are Hamiltonian cycles which visit each vertex exactly once. In CVRP, each vertex $x_i$ has a demand $d_i$ to fulfill, and depot node $x_0$ is introduced for the vehicle to replenish when it runs out of its capacity. The vehicle is constrained to visit vertices except depot strictly once while not exceeding capacity limit.

## 3.2 Policy Learning for Routing Problems

Policy learning is the dominant paradigm in deep reinforcement learning for solving routing problems. Given an instance $\mathcal{X}$, a feasible solution $\boldsymbol{\tau}$ is autoregressively generated through the neural policy:

$$p_\theta(\boldsymbol{\tau}|\mathcal{X}) = p_\theta(\tau_1|\mathcal{X}) \prod_{t=2}^{N} p_\theta(\tau_t|\tau_{1:t-1}, \mathcal{X}), \tag{2}$$

where $\theta$ are the parameters of the policy network, trained by minimizing the expected total cost:

$$\mathcal{L}(\theta|\mathcal{X}) = \mathbb{E}_{p_\theta(\boldsymbol{\tau}|\mathcal{X})}[L(\boldsymbol{\tau})]. \tag{3}$$

REINFORCE [44] is the cornerstone of policy-based reinforcement learning methods, which computes the gradient $\nabla \mathcal{L}(\theta|\mathcal{X})$ in the context of routing problems as the following:

$$\mathbb{E}_{p_\theta(\boldsymbol{\tau}|\mathcal{X})} \left[ (L(\boldsymbol{\tau}) - b(\mathcal{X})) \sum_{t=2}^{N} \nabla \log p_\theta(\tau_t|\tau_{1:t-1}, \mathcal{X}) \right], \tag{4}$$

where $b(\mathcal{X})$ is a baseline for variance reduction.

# 4 Purity Patterns for Generalization

In this section, we first explore the structural patterns closely tied to generalization in optimal solutions of routing problems. We observe that edges surrounded by sparse vertices consistently prevail across different instances. To capture this relationship, we propose the concept of *purity order*, which quantifies the density of vertices around an edge. Based on this, we uncover the Purity Law, an empirical phenomenon that reveals a universal structural principle in optimal solutions of routing problems.

## 4.1 Purity Order in Routing Problems

We first introduce the concept of purity order to formulate the vertex density around each edge.

**Definition 4.1** (Purity Order). *The covering set $N_c$ and purity order $K_p$ of an edge $e_{ij} = (x_i, x_j)$ are defined as:*

$$N_c(e_{ij}) := \{x \in \mathcal{X} \mid (x_i - x)^T \cdot (x_j - x) < 0\}, \tag{5a}$$
$$K_p(e_{ij}) = K_p(x_i, x_j) := |N_c(e_{ij})|. \tag{5b}$$

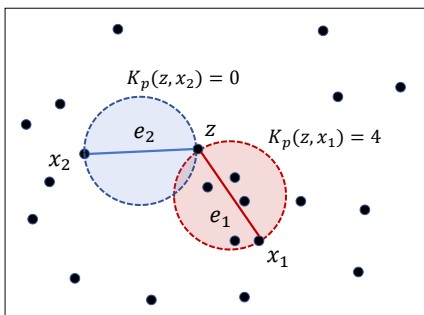

Figure 1: Example of edge purity orders with the same length.

Geometrically, the covering set of an edge corresponds to the vertices lying within a circle, whose diameter is defined by this edge, and the purity order is the cardinality of this set. It is not difficult to obtain that the computational complexity required to calculate the purity order for all nodes is $O(|\mathcal{X}|^2)$. The purity order can reflect the local density of vertices surrounding an edge. A lower purity order indicates a more sparse and "pure" configuration of vertices around the edge. For convenience, we refer to an edge with purity order $k$ as a $k$-order pure edge.

Additionally, we investigate the topological properties of $0$-order pure edges with the lowest redundancy (details and proofs shown in Appendix B). Specifically, we establish the *existence* of $0$-order pure neighbors for any vertex, laying the groundwork for our optimization paradigm. We then prove the *connectivity* of the subgraph formed by $0$-order pure edges, indicating that these edges capture global structural properties. Furthermore, we demonstrate that the polyhedron formed by the $0$-order pure neighbors of any vertex is *convex*, highlighting its structural integrity and stability. This insight suggests the potential for efficient computational methods. We also design the purity order for non-Euclidean problems, which can be referred to Appendix C.

**Intuitive Example Analysis.** Measuring vertex redundancy around an edge, purity order can also capture topological information beyond edge length. Even among edges of equal length, those with lower purity orders typically exhibit more favorable structural properties. As shown in Fig. 1, consider two edges, $e_1$ and $e_2$, formed with vertices $x_1$ and $x_2$ from vertex $z$. Despite of the same length, their purity orders are 4 and 0, respectively. The edge $e_1$ passes through a denser region of vertices, which negatively affects the subsequent connections, making it harder to form a cohesive structure. In contrast, $e_2$, surrounded by sparser vertices, has minimal negative impact on the underlying connectivity of others, suggesting more conducive to a well-structured solution. These observations suggest that solutions with more lower-order pure edges may have a greater potential for optimality.

## 4.2 Purity Law in Optimal Solutions of Routing Problems

We quantitatively investigate the distribution of purity orders in optimal solutions across various instance scales and distributions, leading to the concept of the Purity Law.

> **Purity Law** : The distribution of edges in optimal solutions follows a *negative exponential law* based on their purity orders across various instances.

To validate the Purity Law, we conduct extensive statistical experiments on instances with varying scales and distributions. The following discussions focus on TSP, while the Purity Law also exists in CVRP despite more constraints, with verification and analysis presented in Append. D. Specifically, we create datasets from 84 instance types with scales ranging from 20 to 1000, sampled from four classical distributions. The corresponding optimal solutions are computed using the LKH3 algorithm [17]. A detailed description of the dataset is provided in Append. E.

For each instance type, we calculate the purity order of every edge in the optimal solution and then statistic the proportion $y$ of edges with a given purity order $k$, which is defined as:

$$y(k) = \frac{\sum_{(\tau_i^*, \tau_j^*) \in \tau^*} \mathbb{I}(K_p(\tau_i^*, \tau_j^*) = k)}{N}, \quad k \in [0, N-2], \tag{6}$$

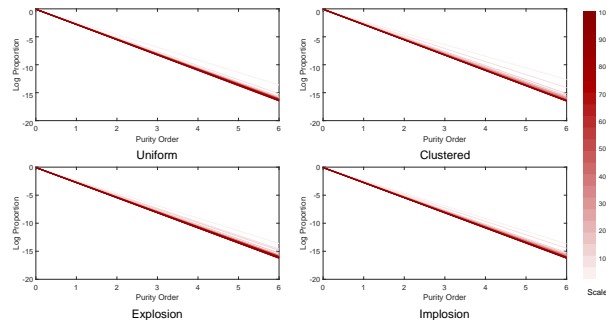

Figure 2: Purity Law curves under varying scales and distributions.

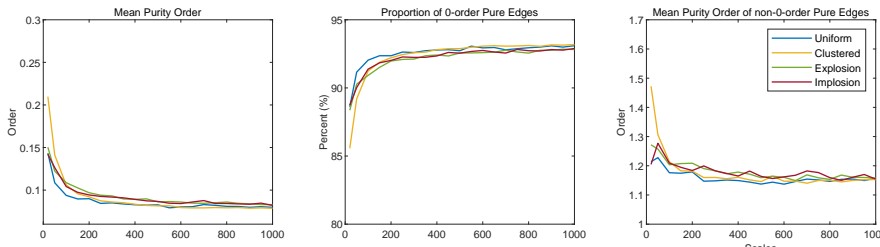

Figure 3: Mean purity order, proportion of $0$-order pure edges, and average order of non-$0$-order pure edges for varying instance.

where $N$ is the length of solution, $(\tau_i^*, \tau_j^*)$ is the edge in the optimal solution $\boldsymbol{\tau}^*$, and $\mathbb{I}(\cdot)$ is the indicator function. Next, we fit the proportions to a negative exponential function:

$$\log y(k) = -\beta k + \log \alpha, \tag{7}$$

and present statistics of fitted parameters $\alpha$, $\beta$, and fitting errors in Table 7 of Appendix F. And all detailed fitting results for each instance type are also available in Appendix F. The remarkably low mean and variance of fitting errors statistically underscore that the negative exponential law reliably and universally applies across different instance scales and distributions.

The negative exponential relationship reveals that purity orders of edges and their proportions in the optimal solution decrease exponentially, confirming our earlier comprehension. To better understand the decay rate, we visualize the fitted curves for different instance types in Fig. 2, where we plot the logarithm of $y$ for clarity. As seen in Fig. 2, the coefficient $\alpha$ is invariant for both scales and distributions, with different curves nearly overlapping at the vertical axis intersection. The decay rate $\beta$ slightly increases with instance scale, indicating that, for larger-scale instances, the proportion of each purity order decreases more rapidly.

**Theorem 4.2.** *For edge $e_{xy}$, assume that $K_p(e_{xy}) = k$ is an positive even number, $z_i \in N_c(e_{xy})$ evenly distributed on both sides of $e_{xy}$, and the probability of each node being selected follows a uniform distribution. Then, the probability of $e_{xy}$ being at the optimal solution holds the following inequality:*

$$\mathbb{P}[e_{xy} \in \tau_{\mathrm{OPT}}] \leq \frac{k!!}{2 \cdot 4^{\frac{k}{2}-1}(k+1)!!} \tag{8}$$

For the random Euclidean routing problem, we theoretically prove that under certain conditions, there is an upper bound about the probability of an edge with purity order $k$ being at the optimal solution, which is shown in Theorem 4.2. The proof of Theorem 4.2 can be referred to Appendix G. Notably, this upper bound decreases monotonically with increasing $k$. This means that edges of higher purity order have a lower probability of forming the optimal solution, consistent with our findings. Furthermore, the negative exponential form of this upper bound indirectly confirms the purity law we propose.

## 4.3 Consistent Dominance of Lowest-order Pure Edges

The rapid decay of the negative exponential function ensures that edges with lower purity orders dominate the optimal solution, while edges with higher purity orders are seldom present. To concretely

illustrate this property, we calculate the mean purity order, the proportion of $0$-order pure edges, and the average order of non-$0$-order pure edges in optimal solutions for each instance type. The results, shown in Fig. 3, highlight the consistent dominance of $0$-order pure edges.

The variations in these metrics exhibit a steady trend across different instance scales and distributions, further reinforcing the universality and stability of the Purity Law. The mean purity order stabilizes around $0.1$, indicating that the optimal tours consistently maintain a quite low level of overall purity order. Furthermore, the proportion of $0$-order pure edges consistently stays around $92\%$, reflecting their dominant presence in optimal solutions. This proportion also aligns with the fitting parameter $\alpha$. For non-$0$-order pure edges, which have a minimal purity order of $1$, the mean purity order remains around $1.15$. This suggests that, even among non-$0$-order pure edges, low-order pure edges are predominant, with higher-order pure edges being quite rare. Notably, all these proportions remain nearly invariant when the instance scale exceeds $300$, underscoring the remarkable and dominant consistency of low-order pure edges. These quantitative characteristics indicate that, compare with the well-known k-nearest prior, Purity Law can reflect additional structural information. Further discussions on connection with k-nearest prior are available in Appendix H.

## 5 Policy Learning Inspired by Purity Patterns

In standard neural learning for routing problems, the total cost, as defined in Eq. (4), is typically the only objective for policy optimization. However, relying solely on this reward signal, neural networks often fail to capture inherent structural patterns present across different instances. Despite efforts to modify the architecture of policy networks and learning modules in previous research, the absence of guidance from generalizable structural principles, limits effective learning of universal patterns, resulting in poor generalization. Fortunately, the widespread and stable presence of purity patterns in optima presents a promising avenue for improving generalization. Motivated by this, we introduce two key concepts–*purity availability* and *purity cost*–to characterize purity patterns for neural learning. By incorporating these into the policy training, we propose PUrity Policy Optimization (PUPO) to encourage alignment with the Purity Law, enhancing generalization in neural solvers.

### 5.1 Purity Availability and Purity Cost

In the Markov model of routing problems, the state at time $t$ consists of the *unvisited vertex set $\mathcal{U}_t$* and the *current partial solution $\boldsymbol{\tau}_t$*, which includes the visited vertex set $\mathcal{V}_t$. The action $a_t$ corresponds to the vertex $\tau_t$ selected for the next visit.

**Definition 5.1** (Purity Availability). *The purity availability $\phi(\cdot)$ of the unvisited vertex set $\mathcal{U}_t$ is defined as the average minimum available purity order, given by:*

$$\phi(\mathcal{U}_t) = \frac{\sum_{x_i \in \mathcal{U}_t} \min_{x_j \in \mathcal{U}_t, j \neq i} K_p(x_i, x_j)}{|\mathcal{U}_t|}, \quad 1 \leq t \leq N. \tag{9}$$

The purity availability $\phi(\mathcal{U}_t)$ measures the potential purity order of future edges that can be formed by the unvisited vertices. We demonstrate that the $\phi$ is supermodular, meaning the absolute value of marginal gain in purity availability diminishes as the set $\mathcal{U}_t$ increases. The proof is in Appendix I.

**Proposition 5.2** (Supermodularity of Purity Availability). *The set function $\phi : 2^{\mathcal{X}} \to \mathbb{R}$, defined on subsets of the finite set $\mathcal{X}$, is supermodular. Specifically, for any subsets $\mathcal{A} \subseteq \mathcal{B} \subseteq \mathcal{X}$ and any vertex $x \in \mathcal{X} \setminus \mathcal{B}$, the following holds:*

$$\phi(\mathcal{A} \cup \{x\}) - \phi(\mathcal{A}) \leq \phi(\mathcal{B} \cup \{x\}) - \phi(\mathcal{B}). \tag{10}$$

This indicates that, as the solution construction progresses, the absolute value of marginal improvement in purity availability becomes more significant on smaller unvisited sets.

**Definition 5.3** (Purity Cost). *The purity cost $C(\mathcal{U}_t, \tau_{t+1})$ for selecting an action $\tau_{t+1}$ at time $t$ is defined as:*

$$\begin{cases} K_p(\tau_t, \tau_{t+1}) + \phi(\mathcal{U}_{t+1}) - \phi(\mathcal{U}_t), & \text{for } t < N, \\ K_p(\tau_N, \tau_1), & \text{for } t = N. \end{cases} \tag{11}$$

The purity cost $C(\mathcal{U}_t, \tau_{t+1})$ includes both the purity order of the new edge formed by $\tau_t$ and $\tau_{t+1}$, and the difference in purity availability before and after the action $\tau_{t+1}$. This formulation captures

both the contribution of action $\tau_{t+1}$ to the purity order of the current partial solution and its impact on the purity potential of the remaining unvisited vertices.

## 5.2 Policy Optimization with Purity Weightings

To improve generalization, we integrate purity costs into policy optimization process, aligning model with the Purity Law. This encourages the learning of structural patterns that are consistent across varying instances. To assess the potential of the current state-action pair in fostering a low-purity structure for future solution construction, we introduce the purity weighting based on purity costs.

**Definition 5.4** (Purity Weightings). *Let $\delta$ be a discount factor. The purity weighting $W(\mathcal{U}_t, \tau_{t+1})$ is defined as:*

$$W(\mathcal{U}_t, \tau_{t+1}) = 1 + \sum_{j=t}^{N} \delta^{j-t} C(\mathcal{U}_j, \tau_{j+1}). \tag{12}$$

Building on this characterization, we propose the *purity policy gradient* $\nabla \mathcal{L}_{PUPO}(\theta|\mathcal{X})$ as follow:

$$\mathbb{E}_{p_\theta(\boldsymbol{\tau}|\mathcal{X})} \left[ (L(\boldsymbol{\tau}) - b(\mathcal{X})) \sum_{t=2}^{N} W(\mathcal{U}_{t-1}, \tau_t) \nabla \log p_\theta(\tau_t | \tau_{1:t-1}, \mathcal{X}) \right], \tag{13}$$

The complete description of the PUPO framework is presented in Appendix J. And we also provide an theoretical analysis on optimization error of PUPO in Appendix K. PUPO establishes upon the REINFORCE algorithm with a POMO baseline, incorporating generalizable purity patterns into the modified policy gradient. At each stage of the solution construction, PUPO employs a discounted purity cost that reflects the purity potential of both the current partial solution and the unvisited vertex set, encouraging the model to favor actions with lower purity orders during training. This approach helps the model to learn consistent, cross-instance patterns that align with the Purity Law, thereby enhancing its generalization capability. Notably, PUPO is flexible and can be integrated with various popular constructive neural solvers, without any alterations to the network architecture. Further discussion on the derivation process and explanation of PUPO are provided in Appendix L.

## 6 Numerical Experiments

To validate the effect of PUPO on enhancing generalization, we compare the performance of both vanilla policy optimization and PUPO across several state-of-the-art constructive neural solvers. Additionally, we conduct a comprehensive evaluation of generalization performance and purity metrics on well-known public datasets.

### 6.1 Experimental Setups

**Dataset.** To ensure the adequacy of the experiments, we validate the model's performance on two categories of datasets. The randomly generated dataset used in this paper is the same as that in INViT [10], which is widely adopted to testify existing DRL approach. The real-world dataset we used is TSPLIB [39] and CVRPLIB [40]. More details about dataset we used are presented in Appendix N.

**Comparison Methods.** Although PUPO can be integrated with any DRL-based neural constructive model, we select four representative SOTA methods with high recognition for our experiments. This section mainly present the results on two latest and advanced neural solvers ELG [12] and INViT [10] We also conduct experiments on two classical neural solvers POMO [26] and PF [18], whose results can be seen in Appendix M. For each method, we conduct training on scales of 50 and 100.

**Evaluation Metrics.** We report three widely adopted performance metrics to evaluate the generalizability of each comparison method, including the average total cost of solutions, the average gap to the optimal solutions, and the average solving time. The gap characterizes the relative difference between the output solution of neural models and the optimal solution, which is calculated as the following:

$$gap = \frac{L(\boldsymbol{\tau}^{model}) - L(\boldsymbol{\tau}^{opt})}{L(\boldsymbol{\tau}^{opt})} \times 100\% \tag{14}$$

We also present three metrics to reflect the ability of the model to perceive purity information: the average purity order (APO (all) for short), the proportion of 0-order pure edges (Prop-0 (%) for short), and the average order of non-0-order pure edges (APO (non-0) for short) for each solution.

**Experimental Settings.** All the numerical experiments are implemented on an NVIDIA GeForce RTX 3090 GPU with 24 GB of memory, paired with a 12th Gen Intel(R) Core(TM) i9-12900 CPU. We train each model using the vanilla method and PUPO, respectively. In each training paradigm, we use the original network architectures and hyper-parameters provided in the source code, *without any modifications to modules*. Due to the different magnitude in the policy gradient between the two, we only adjust the learning rates during PUPO. Details about the learning rates of each model are available in Appendix O. All models are trained on random instances from a uniform distribution.

## 6.2 Generalization Performance Analysis

Table 1: The experimental results of average gap (%) on the randomly generated TSP dataset with different distributions and scales after training each model using both the vanilla and PUPO methods, where - means out of memory, bold formatting represents superior results, Blue highlights the relative decrease ratio (%) in the gap, while Red indicates its relative increase ratio (%).

| Instance | ELG-50 Vanilla | PUPO | | ELG-100 Vanilla | PUPO | | INViT-50 Vanilla | PUPO | | INViT-100 Vanilla | PUPO | |
|---|---|---|---|---|---|---|---|---|---|---|---|---|
| U-100 | **6.60** | 7.10 | ↑7.62 | **5.86** | 6.39 | ↑9.13 | **2.14** | 2.17 | ↑1.59 | 2.61 | **2.35** | ↓9.92 |
| U-1000 | 20.81 | **19.47** | ↓6.40 | 17.95 | **17.49** | ↓2.58 | 7.27 | **7.17** | ↓1.43 | 7.41 | **6.26** | ↓15.45 |
| U-5000 | 31.50 | **26.84** | ↓14.78 | 29.22 | **23.77** | ↓18.66 | 9.41 | **9.09** | ↓3.37 | 9.39 | **8.05** | ↓14.25 |
| U-10000 | - | - | | - | - | | 7.89 | **7.63** | ↓3.39 | 7.51 | **6.25** | ↓16.74 |
| C-100 | 10.05 | **8.98** | ↓10.60 | 11.50 | **9.37** | ↓18.45 | 2.90 | **2.86** | ↓1.55 | 3.59 | **3.22** | ↓10.23 |
| C-1000 | 27.14 | **24.13** | ↓11.09 | 27.06 | **21.13** | ↓21.90 | 8.44 | **8.00** | ↓5.25 | 8.37 | **7.20** | ↓13.93 |
| C-5000 | 38.89 | **35.63** | ↓8.39 | 40.44 | **29.29** | ↓27.58 | 9.90 | **9.45** | ↓4.51 | 9.65 | **8.55** | ↓11.35 |
| C-10000 | - | - | | - | - | | 10.85 | **10.09** | ↓6.92 | 10.47 | **9.11** | ↓13.00 |
| E-100 | **6.85** | 7.23 | ↑5.64 | 7.28 | **7.19** | ↓1.22 | 2.15 | **2.11** | ↓2.23 | 2.70 | **2.41** | ↓10.88 |
| E-1000 | 24.55 | **23.77** | ↓3.21 | 24.01 | **21.57** | ↓10.16 | 9.61 | **9.18** | ↓4.39 | 9.61 | **8.77** | ↓8.76 |
| E-5000 | 35.67 | **28.88** | ↓19.03 | 34.77 | **26.26** | ↓24.47 | 12.44 | **11.65** | ↓6.36 | 11.45 | **10.41** | ↓9.03 |
| E-10000 | - | - | | - | - | | 11.22 | **10.61** | ↓5.44 | 10.96 | **9.62** | ↓12.24 |
| I-100 | **6.66** | 7.23 | ↑8.69 | 6.64 | **6.60** | ↓0.56 | 2.44 | **2.38** | ↓2.22 | 2.77 | **2.62** | ↓5.46 |
| I-1000 | 21.30 | **20.54** | ↓3.56 | 18.62 | **18.52** | ↓0.57 | 7.56 | **7.50** | ↓0.70 | 7.81 | **6.90** | ↓11.65 |
| I-5000 | 30.40 | **27.64** | ↓9.07 | 30.70 | **24.64** | ↓19.76 | 9.20 | **8.86** | ↓3.60 | 9.44 | **8.96** | ↓5.14 |
| I-10000 | - | - | | - | - | | 8.36 | **7.68** | ↓8.10 | 8.27 | **6.78** | ↓18.05 |

Table 2: The experimental results of average gap (%) on the randomly generated CVRP dataset with different distributions and scales after training each model using both the vanilla and PUPO methods.

| Instance | ELG-50 Vanilla | PUPO | | ELG-100 Vanilla | PUPO | | INViT-50 Vanilla | PUPO | | INViT-100 Vanilla | PUPO | |
|---|---|---|---|---|---|---|---|---|---|---|---|---|
| U-50 | **7.45** | 8.15 | ↑9.45 | **8.32** | 9.09 | ↑9.36 | **4.25** | 4.58 | ↑7.83 | **4.75** | 5.06 | ↑6.42 |
| U-500 | 15.48 | **10.97** | ↓29.14 | 8.91 | **8.84** | ↓0.73 | 10.75 | **9.83** | ↓8.55 | 9.67 | **8.37** | ↓13.50 |
| U-5000 | 22.75 | **8.31** | ↓63.47 | 5.97 | **5.14** | ↓13.85 | 9.67 | **7.26** | ↓24.98 | 7.90 | **4.72** | ↓40.18 |
| C-50 | 7.45 | **7.28** | ↓2.36 | **7.40** | 8.05 | ↑8.82 | **4.50** | 4.84 | ↑7.58 | **4.64** | 4.97 | ↑7.20 |
| C-500 | 16.47 | **10.21** | ↓37.98 | 8.82 | **7.92** | ↓10.18 | 9.79 | **8.93** | ↓8.79 | 9.22 | **7.51** | ↓18.56 |
| C-5000 | 20.45 | **8.72** | ↓57.36 | 7.10 | **5.43** | ↓23.50 | 8.49 | **7.07** | ↓16.72 | 7.66 | **4.51** | ↓41.07 |
| E-50 | **7.58** | 8.13 | ↑7.32 | **8.57** | 9.15 | ↑6.79 | **4.56** | 4.94 | ↑8.52 | **4.88** | 5.37 | ↑9.98 |
| E-500 | 15.86 | **11.12** | ↓29.88 | 9.32 | **8.68** | ↓6.93 | 10.48 | **9.82** | ↓6.26 | 9.82 | **8.52** | ↓13.27 |
| E-5000 | 27.10 | **8.52** | ↓68.58 | 8.26 | **6.33** | ↓23.36 | 9.64 | **7.71** | ↓20.08 | 8.47 | **5.17** | ↓39.03 |
| I-50 | **7.77** | 8.22 | ↑5.85 | **8.34** | 8.67 | ↑3.94 | **4.41** | 4.77 | ↑8.26 | **4.78** | 5.08 | ↑6.29 |
| I-500 | 15.76 | **10.83** | ↓31.29 | 8.97 | **8.83** | ↓1.53 | 10.15 | **9.49** | ↓6.48 | 9.34 | **8.28** | ↓11.35 |
| I-5000 | 19.43 | **7.05** | ↓63.73 | 6.01 | **5.24** | ↓12.91 | 8.23 | **6.87** | ↓16.48 | 7.35 | **4.58** | ↓37.66 |

**Performance on Randomly Generated Dataset.** Table 1 and 2 presents the performance on the randomly generated TSP and CVRP dataset, trained using both the vanilla and PUPO methods. The experimental results for total cost, statistics of gap and solving time are provided in Appendix P. It can be observed that PUPO training enhances the performance across nearly all instance types on both TSP and CVRP. Notably, the PUPO-trained INViT-100 accomplishes a gap of 4.51% on C-5000 of CVRP, whereas the Vanilla-trained model can only reach a gap of 7.66%.

From the perspective of test scale, PUPO training significantly improves model generalization performance on larger instances. For example, INViT-100 achieves a 18.05% relative enhancement on I-10000 of TSP, 40.18% and 41.07% on U-5000 and C-5000 of CVRP. And ELG-50 shows significantly 68.58% and 63.73% gain on E-5000 and I-5000 of CVRP. On scales closer to the training scale, diverse phenomena can be observed. INViT-100 accomplishes an 9.92% uplift on U-100 of TSP, while most other PUPO-trained solvers experience negative performance changes on the

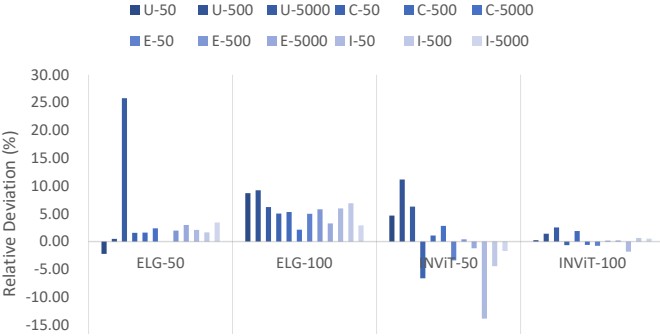

Figure 4: The bar chart of relative deviations (%) in solving time between Vanilla- and PUPO-trained models, where positive values indicate that PUPO-trained models hold shorter solving times.

same scale. This suggests that PUPO may involve in a trade-off, sacrificing accuracy on smaller-scale instances while learning patterns more suited to larger scales. This trade-off is linked to the implicit regularization feature of PUPO, which is discussed in the subsequent part. From a distributional perspective, PUPO leads to positive average relative improvements across all distributions, indicating an enhancement in generalizability with respect to distributions.

**Performance on Real-world Dataset.** Table 3 presents the performance on TSPLIB, further demonstrating that PUPO enhances the generalization ability on real-world data. For example, PUPO-trained ELG-50 achieves 31.6% relative enhancement on CVRPLIB $501 \sim 1000$. Similar to the experimental results on random datasets, PUPO training leads to better performance on large-scale instances, but there are still some unimproved instances at small scale.

Table 3: Performance of average gap (%) on TSPLIB and CVRPLIB after training each model using both the vanilla and PUPO methods.

| Instance | Vanilla | PUPO | | Vanilla | PUPO | |
|---|---|---|---|---|---|---|
| | | ELG-50 | | | ELG-100 | |
| TSPLIB $1\sim100$ | **4.25** | 4.57 | ↑ 7.57 | 4.91 | **4.90** | ↓ 0.10 |
| TSPLIB $101\sim1000$ | 10.81 | **10.21** | ↓ 5.50 | **9.47** | 10.16 | ↑ 7.27 |
| TSPLIB $1001\sim5000$ | 22.88 | **20.24** | ↓ 11.55 | 23.45 | **18.69** | ↓ 20.27 |
| CVRPLIB $1\sim200$ | 9.84 | **9.52** | ↓ 3.32 | **8.30** | 8.62 | ↑ 3.81 |
| CVRPLIB $201\sim500$ | 13.11 | **11.26** | ↓ 14.13 | 9.73 | **9.55** | ↓ 1.90 |
| CVRPLIB $501\sim1000$ | 16.25 | **11.12** | ↓ 31.60 | 10.18 | **9.96** | ↓ 2.18 |
| | | INViT-50 | | | INViT-100 | |
| TSPLIB $1\sim100$ | **1.71** | 1.84 | ↑ 7.78 | 2.43 | **2.16** | ↓ 10.91 |
| TSPLIB $101\sim1000$ | 4.97 | **4.67** | ↓ 6.01 | 5.60 | **4.97** | ↓ 11.15 |
| TSPLIB $1001\sim5000$ | 9.65 | **8.49** | ↓ 12.00 | 9.32 | **8.68** | ↓ 6.87 |
| CVRPLIB $1\sim200$ | **8.93** | 9.00 | ↑ 0.78 | **8.72** | 9.57 | ↑ 9.84 |
| CVRPLIB $201\sim500$ | 12.61 | **12.31** | ↓ 2.36 | 12.05 | **11.34** | ↓ 5.88 |
| CVRPLIB $501\sim1000$ | 13.31 | **12.59** | ↓ 5.45 | 12.31 | **11.28** | ↓ 8.37 |

## 6.3 Computational Efficiency Analysis

Figure 4 illustrates the relative deviations in solving time between Vanilla-trained and PUPO-trained models on CVRP, where positive values indicate that PUPO-trained models hold shorter solving times. Following the figure, most of the relative deviations are within 10%, suggesting negligible differences between the two, even the PUPO-trained models tend to solve faster in most cases. The possible reason may derive from that PUPO acts as an implicit regularization, which may have a positive impact on the computation of the model during the forward inference phase. Since PUPO only provides guidance during the training phase, it does not introduce additional time overhead during inference. Due to tensorizable computation of PUPO, training time does not increase greatly.

## 6.4 Learning Mechanisms Analysis

**Purity Perception.** To further explore the role of PUPO during the training process, Table 4 presents three metrics that evaluate the purity of tours obtained by INViT-100. Numerical results of purity

metrics for all models are available in Appendix P. The table demonstrates that PUPO-trained models consistently generate solutions with more outstanding purity.

Table 4: The experimental results of Vanilla- and PUPO-trained model on three purity evaluation metrics, where bold formatting represents more pure results.

| | Prop-0 (%) | | APO (all) | | APO (non-0) | |
| | Vanilla | PUPO | Vanilla | PUPO | Vanilla | PUPO |
|---|---|---|---|---|---|---|
| 100 | 80.02 | **80.53** | 0.13 | **0.12** | 1.16 | **1.12** |
| 1000 | 85.46 | **86.48** | 0.23 | **0.19** | 1.81 | **1.66** |
| 5000 | 43.36 | **43.95** | 0.31 | **0.24** | 2.52 | **2.10** |
| 10000 | 87.17 | **88.27** | **0.54** | 0.55 | **4.28** | 4.86 |
| Uniform | 74.36 | **75.23** | 0.17 | **0.14** | 1.44 | **1.33** |
| Clustered | 73.82 | **74.65** | 0.34 | **0.33** | **2.76** | 3.02 |
| Explosion | 73.91 | **74.71** | 0.48 | **0.42** | 3.85 | **3.64** |
| Implosion | 73.91 | **74.64** | 0.21 | **0.20** | **1.73** | 1.74 |

Compared to vanilla training, PUPO enhances the Prop-0 metric across all types, and it lead to a reduction in the two kinds of APO metric across nearly all types. Notably, the largest average improvement in Prop-0 occurs at the scale of 10,000, with a 1.1% increase, which also corresponds to the scale where INViT-100 achieves its most promoted generalization performance. These results suggest that PUPO facilitates the emergence of Purity Law during training, leading to enhanced model generalization.

**Implicit Regularization.** Furthermore, we calculate the sum of the Frobenius norm of the parameter matrices for each model after different training, as presented in Table 5. It can be observed that PUPO training reduces the sum of the parameter norms, acting as an implicit regularization mechanism. In contrast, we also perform explicit regularization training by directly incorporating the sum of the norms into the reward function, but it does not result in improved generalizability. This suggests that PUPO effectively mitigates overfitting on the specific training set, encouraging the model to learn more generalizable structural patterns.

Table 5: The sum of Frobenius norm of the parameter matrices for each model after different training, where bold formatting represents lower value.

| | ELG-50 | ELG-100 | INViT-50 | INViT-100 |
|---|---|---|---|---|
| Vanilla | 336.44 | 336.24 | 430.83 | 429.78 |
| PUPO | **335.8** | **335.67** | **425.26** | **425.19** |

## 7 Conclusion

In this paper, we reveal Purity Law, a fundamental structural principle for optima of routing problems, which defines that edge prevalence grows exponentially with the sparsity of surrounding vertices. And we propose Purity Policy Optimization (PUPO) to explicitly promote alignment with Purity Law during the solution construction process. Extensive experiments demonstrate that PUPO can be seamlessly integrated with popular neural solvers, significantly enhancing their generalization performance without incurring additional computational overhead during inference. Purity Law provides a novel perspective on routing problems, revealing a systematic bias toward local sparsity in global optima validated across diverse instances statistically. While improving large-scale generalization, maintaining high accuracy for small-scale scenarios is still a limitation PUPO needs to improve. In future work, we plan to explore more effective training methods for leveraging Purity Law, and design network architectures that integrate Purity Law to promote the perception to it, thereby achieving greater generalizability.

## Acknowledgments

This paper is supported by the National Key R&D Program of China project (2021YFA1000403), and the National Natural Science Foundation of China (Nos. 12431012, U23B2012).

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

# A  More Discussion on Related Works.

With the rapid progress in deep learning, various neural approaches to combinatorial optimization have emerged. Overviews of these methods can be found in Guo et al. [14] and Bengio et al. [5].

**Classical Neural Solvers.** According to the way the solutions are generated, neural approaches can generally be divided into two classes: learning to directly construct and learning to iteratively improve solutions. For constructive methods, Vinyals et al. [43] introduce the Pointer Network, which solves routing problems end-to-end utilizing Recurrent Neural Networks to encode vertex embeddings, trained via supervised learning. Kool et al. [24] first introduce the Transformer architecture [41] to solve routing problems. Kwon et al. [26] further propose the policy optimization with multiple optima (POMO), which improves the performance by exploiting solution symmetries. Meanwhile, some works based on graph learning [21, 20] also show potential to solve routing problems. Pointer-former [18] enhances memory efficiency by adopting a reversible residual network in the encoder and a multi-pointer network in the decoder. For iterative methods, DRL-2opt [7] trains a DRL policy to select appropriate 2-opt operators for refining solutions. Ma et al. [33] design dual-aspect collaborative Transformer to learn embeddings for the node and positional features separately, iteratively solving routing problems. GCN+MCTS [11] integrates graph decomposition and Monte Carlo Tree Search to handle large-scale instances effectively but time-consuming.

Next, we will supplement more related work on improving generalization of neural routing problem solvers. AMDKD [6] introduces knowledge distillation to tackle the cross-distribution generalization concerns in the routing problem. Specifically, it leverages knowledge from teachers trained on exemplar distributions to yield a generalist student model. Also along with knowledge distillation, PDAM [46] adopts curriculum learning to train TSP samples in increasing order of their problem size and progressively distilling high-level knowledge from small models to large models via a distillation loss. SCA [23] designs a plugged network to mix scale information into the original representation vector, which can help the pre-trained model adapt the policy to larger-scale tasks. By studying adversarial robustness, Geisler [13] derives perturbation models for SAT and TSP to enhance the expressiveness of the model with perturbations. Zhou [47] proposes a generic meta-learning framework to enhance the ability of the initialized model to fast adapt to new tasks during inference and conduct extensive experiments on VRPs. Sahil [34] Formalizes solving a CO problem over a given instance distribution as a separate learning task and investigates meta-learning to optimize the capacity of the model to adapt to new tasks. Reformulating the Markov Decision Process of the solution construction in combinatorial optimization problems, BQ-NCO [8] proposes a novel Bisimulation Quotienting method for generalizable neural solver. GDMF [29] proposes an divide-and-conquer framework via fusing multi-level feature for large-scale TSP.

# B  Topological Properties of $0$-order Pure Edges

As the class of edges with the lowest degree of redundancy, $0$-order pure edges are demonstrated by the following propositions to possess a series of favorable topological properties.

**Proposition B.1.** *Given an instance $\mathcal{X}$, for any vertex $x \in \mathcal{X}$, there exists at least one vertex that can form a 0-order pure edge with $x$.*

*Proof.* We prove the proposition by contradiction.

Assume that for any $x \in \mathcal{X}$, we choose

$$y(x) := \arg\min \|x - y\|^2.$$

Since no vertex can form a $0$-order pure edge with $x$, we have that

$$K_p(x, y(x)) > 0, \quad \forall x \in \mathcal{X}.$$

Therefore, there must exist a point $z(x)$ such that

$$(x - z(x))^T (y(x) - z(x)) < 0.$$

Then we can derive that

$$\|x - y(x)\|^2$$
$$= \|x - z(x) + z(x) - y(x)\|^2$$
$$= \|x - z(x)\|^2 + \|z(x) - y(x)\|^2 - 2(x - z(x))^T (y(x) - z(x))$$
$$\geq \|x - z(x)\|^2 + \|z(x) - y(x)\|^2$$
$$\geq \|x - z(x)\|^2,$$

which means that the distance between $z(x)$ and $x$ is smaller than the distance between $y(x)$ and $x$. Therefore, it contradicts the assumption that $y(x)$ is the nearest neighbor of $x$. □

**Proposition B.2.** *Given an instance $\mathcal{X}$, the subgraph $G_0 = (\mathcal{X}, \mathcal{E}_0)$ is connected, where $\mathcal{E}_0$ is the edge set of all 0-order pure edges.*

*Proof.* We prove the proposition by contradiction.

Suppose the subgraph $G_0$ is not connected. Without loss of generality, assume that $G_0$ has two connected components, denoted as $F_1$ and $F_2$. Let $f_1 \in F_1$ and $f_2 \in F_2$ be such that

$$\text{dist}(f_1, f_2) = \text{dist}(F_1, F_2) = \min_{a \in F_1} \min_{b \in F_2} \text{dist}(a, b).$$

Since $e_{f_1 f_2} \notin G_0$, we have that

$$K_p(f_1, f_2) > 0.$$

Therefore, there must exist a point $f_3$ such that the following inequality holds:

$$(f_1 - f_3)^T (f_2 - f_3) < 0,$$

which means that the distance between $f_3$ and $f_1$ is smaller than the distance between $f_2$ and $f_1$. Regardless of whether $f_3 \in F_1$ or $f_3 \in F_2$, this leads to a contradiction with the assumption that the distance between $f_1$ and $f_2$ is the shortest distance between $F_1$ and $F_2$.

□

**Proposition B.3.** *Given an instance $\mathcal{X}$, for any vertex $x \in \mathcal{X}$, the polyhedron formed by its 0-order pure neighbors $D_0^x$ is convex.*

*Proof.* Consider any three adjacent 0-order pure neighbors $A$, $B$, and $C$ of point $X$. We have that

$$\angle ABC = \angle ABX + \angle XBC.$$

Since $A$ and $C$ are 0-order pure neighbors of $X$, it follows that

$$\angle ABX, \angle XBC < \frac{\pi}{2},$$

and thus $\angle ABC < \pi$.

Due to the arbitrariness of $A$, $B$, and $C$, the polyhedron formed by its 0-order pure neighbors $D_0^x$ is convex. □

Proposition B.1 establishes the existence of 0-order pure neighbors for any point, providing a foundation for subsequent analysis. While proposition B.2 demonstrates that the subgraph formed by 0-order pure edges possesses overall graph structural properties. And proposition B.3 describes the intrinsic topological features of the 0-order neighbor set for any given point. The three propositions above characterize the mathematical properties of purity order theoretically.

## C    Purity Order Form for Non-Euclidean Problems

In non-Euclidean problems, there is no longer direct connectivity between vertices, and the metric is given a priori by the edge cost matrix. Given $G = (V, E)$, we define purity order $K_p(e_{ij})$ in non-Euclidean routing problems as follow:

$$K_p(e_{ij}) = \frac{\max_{m_i \in M_i} |\xi_{im_i}| + \max_{m_j \in M_j} |\xi_{jm_j}|}{2}, \tag{15}$$

where $M_i = \{m | argmax_{m, l(\xi_{im_i}) \leq \frac{l(e_{ij})}{2}} l(\xi_{im_i})\}$, $\xi_{im_i}$ is the shortest path from vertex $v_i$ to $v_{m_i}$

To understand it intuitively, we first need to find the shortest path starting from the two endpoints, which does not exceed half the length of the edge and has the largest length, and then define the purity order of the two end points as the average of the maximum number of nodes in the shortest path. We take an instance of the non-Euclidean TSP dataset of size 50 in [27] as an example and calculate the pure order according to the above definition. The purity order distribution is as follows:

## D    Verification of Purity Law on CVRP

Although different routing problems face distinct constraints, the common goal of VRPs is to minimize the sum of the total tour cost, which is typically characterized by the spatial relationships between nodes. The Purity Law describes a universal local structure of spatial relationships, which are highly relevant to the common goal of VRPs. We also verify the Purity Law on CVRP. Using the dataset from the original paper of INViT [10], we computed the proportion (%) of edges with purity orders $0 \sim 10$ in the optima. The result is shown in the table6. It can be observed that the negative exponential law still obviously holds. And lowest-order pure edges still keep dominance in the optimal solutions, while the concentration is slightly reduced due to capacity constraints.

Table 6: The verification result of Purity Law on CVRP.

| Purity Order | CVRP-50 | | | | CVRP-500 | | | |
|---|---|---|---|---|---|---|---|---|
| | Uniform | Clustered | Explosion | Implosion | Uniform | Clustered | Explosion | Implosion |
| 0 | 78.3 | 76.5 | 78.2 | 78.4 | 66.6 | 65.8 | 67.3 | 66.8 |
| 1 | 13.2 | 12.9 | 13.3 | 13.2 | 11.3 | 11.2 | 11.2 | 11.3 |
| 2 | 5.5 | 5.6 | 5.5 | 5.3 | 4.3 | 4.4 | 4.3 | 4.4 |
| 3 | 3.1 | 3.1 | 3.0 | 3.1 | 2.4 | 2.5 | 2.3 | 2.4 |
| 4 | 2.1 | 2.2 | 2.0 | 2.1 | 1.6 | 1.6 | 1.5 | 1.6 |
| 5 | 1.5 | 1.6 | 1.6 | 1.5 | 1.1 | 1.2 | 1.1 | 1.1 |
| 6 | 1.2 | 1.3 | 1.3 | 1.2 | 0.9 | 1.0 | 0.9 | 0.9 |
| 7 | 1.0 | 1.0 | 1.0 | 1.0 | 0.7 | 0.7 | 0.7 | 0.7 |
| 8 | 0.9 | 0.9 | 0.8 | 0.9 | 0.7 | 0.7 | 0.6 | 0.6 |
| 9 | 0.7 | 0.7 | 0.7 | 0.8 | 0.5 | 0.6 | 0.5 | 0.6 |
| 10 | 0.7 | 0.6 | 0.6 | 0.6 | 0.5 | 0.5 | 0.5 | 0.5 |

## E    Detailed Description of Statistical Dataset

To construct the dataset in statistical experiments, we first generate 21 different scales within the range of 20 to 1000, with intervals of 50 except 20. For each scale, we consider four widely recognized classical distributions, which is uniform, cluster, explosion and implosion, resulting in 84 different instance types totally. For instance types with scales under 500, 256 instances are randomly sampled to form the dataset, while for those with sizes of 500 or above, the number of instances is reduced to 128, which is due to the fact that optimal solutions of large-scale instances require an excessive amount of time. The optimal solutions for all instances are solved by LKH-3 [16, 17], which is the SOTA heuristic capable of producing optimal solutions even for large-scale instances.

## F    Detailed Fitting Result

The mean and variance of fitted parameters $\alpha$ and $\beta$, along with fitting errors are shown in Table 7. And the detailed values of the fitting error, $\alpha$ and $\beta$ for each instance type are listed in Table 8. According to the table, the low mean and variance of the fitting errors demonstrate the reliability and university of the fitting results across different instance scales and distributions.

Table 7: Fitting results of the exponential function in Eq. (7).

| | FITTING ERROR | $\alpha$ | $\beta$ |
|---|---|---|---|
| MEAN | 2.23E-05 | 0.92 | 2.63 |
| VARIANCE | 8.45E-10 | 1.57E-04 | 1.49E-02 |

Table 8: The fitting errors, coefficients $\alpha$ and $\beta$ in Sec. 4.2

| | Fitting Error | | | | $\alpha$ | | | | $\beta$ | | | |
|---|---|---|---|---|---|---|---|---|---|---|---|---|
| | Uniform | Clustered | Explosion | Implosion | Uniform | Clustered | Explosion | Implosion | Uniform | Clustered | Explosion | Implosion |
| 20 | 6.58E-05 | 2.45E-04 | 1.01E-04 | 5.71E-05 | 0.8860 | 0.8554 | 0.8835 | 0.8872 | 2.2514 | 2.0800 | 2.2483 | 2.2630 |
| 50 | 5.42E-05 | 7.90E-05 | 3.96E-05 | 5.05E-05 | 0.9115 | 0.8919 | 0.9027 | 0.9002 | 2.5166 | 2.3295 | 2.4152 | 2.3923 |
| 100 | 2.41E-05 | 3.75E-05 | 1.85E-05 | 3.61E-05 | 0.9204 | 0.9124 | 0.9091 | 0.9137 | 2.5974 | 2.5133 | 2.4567 | 2.5286 |
| 150 | 2.54E-05 | 2.57E-05 | 2.96E-05 | 1.78E-05 | 0.9236 | 0.9188 | 0.9150 | 0.9184 | 2.6410 | 2.5782 | 2.5412 | 2.5681 |
| 200 | 2.18E-05 | 2.20E-05 | 2.62E-05 | 1.63E-05 | 0.9236 | 0.9221 | 0.9197 | 0.9202 | 2.6384 | 2.6192 | 2.6002 | 2.5887 |
| 250 | 1.18E-05 | 1.35E-05 | 2.38E-05 | 2.42E-05 | 0.9263 | 0.9245 | 0.9209 | 0.9227 | 2.6568 | 2.6366 | 2.6066 | 2.6337 |
| 300 | 1.26E-05 | 1.49E-05 | 2.06E-05 | 2.08E-05 | 0.9259 | 0.9258 | 0.9211 | 0.9224 | 2.6535 | 2.6578 | 2.6057 | 2.6201 |
| 350 | 1.39E-05 | 1.40E-05 | 1.61E-05 | 1.71E-05 | 0.9273 | 0.9263 | 0.9236 | 0.9224 | 2.6762 | 2.6621 | 2.6318 | 2.6166 |
| 400 | 1.33E-05 | 1.54E-05 | 1.80E-05 | 1.05E-05 | 0.9278 | 0.9280 | 0.9241 | 0.9235 | 2.6814 | 2.6918 | 2.6430 | 2.6206 |
| 450 | 1.04E-05 | 1.12E-05 | 1.61E-05 | 1.52E-05 | 0.9280 | 0.9288 | 0.9234 | 0.9260 | 2.6805 | 2.6947 | 2.6289 | 2.6633 |
| 500 | 8.38E-06 | 1.15E-05 | 1.18E-05 | 1.28E-05 | 0.9273 | 0.9288 | 0.9256 | 0.9255 | 2.6634 | 2.6936 | 2.6515 | 2.6536 |
| 550 | 1.04E-05 | 1.49E-05 | 1.94E-05 | 1.31E-05 | 0.9306 | 0.9298 | 0.9257 | 0.9267 | 2.7190 | 2.7186 | 2.6615 | 2.6673 |
| 600 | 7.55E-06 | 1.08E-05 | 1.48E-05 | 1.26E-05 | 0.9294 | 0.9306 | 0.9259 | 0.9275 | 2.6931 | 2.7207 | 2.6591 | 2.6803 |
| 650 | 1.13E-05 | 1.37E-05 | 9.63E-06 | 1.79E-05 | 0.9297 | 0.9310 | 0.9262 | 0.9263 | 2.7058 | 2.7293 | 2.6535 | 2.6689 |
| 700 | 1.34E-05 | 1.02E-05 | 1.40E-05 | 1.77E-05 | 0.9280 | 0.9306 | 0.9277 | 0.9256 | 2.6856 | 2.7175 | 2.6877 | 2.6563 |
| 750 | 1.22E-05 | 1.19E-05 | 1.50E-05 | 1.26E-05 | 0.9287 | 0.9308 | 0.9264 | 0.9282 | 2.6959 | 2.7251 | 2.6658 | 2.6930 |
| 800 | 1.22E-05 | 1.29E-05 | 9.97E-06 | 1.19E-05 | 0.9294 | 0.9311 | 0.9255 | 0.9273 | 2.7040 | 2.7323 | 2.6459 | 2.6750 |
| 850 | 1.30E-05 | 1.12E-05 | 1.80E-05 | 1.33E-05 | 0.9300 | 0.9305 | 0.9276 | 0.9271 | 2.7151 | 2.7184 | 2.6909 | 2.6707 |
| 900 | 1.33E-05 | 1.03E-05 | 1.08E-05 | 1.46E-05 | 0.9309 | 0.9316 | 0.9276 | 0.9281 | 2.7296 | 2.7357 | 2.6790 | 2.6906 |
| 950 | 1.33E-05 | 1.38E-05 | 1.44E-05 | 1.41E-05 | 0.9299 | 0.9314 | 0.9284 | 0.9277 | 2.7132 | 2.7376 | 2.6951 | 2.6855 |
| 1000 | 1.51E-05 | 1.21E-05 | 1.26E-05 | 1.24E-05 | 0.9310 | 0.9318 | 0.9282 | 0.9289 | 2.7340 | 2.7413 | 2.6890 | 2.6978 |

# G  Proof of the upper bound

**Theorem G.1.** *For edge $e_{xy}$, assume that $K_p(e_{xy}) = k$ is an positive even number, $z_i \in N_c(e_{xy})$ evenly distributed on both sides of $e_{xy}$, and the probability of each node being selected follows a uniform distribution. Then, the probability of $e_{xy}$ being at the optimal solution holds the following inequality:*

$$\mathbb{P}[e_{xy} \in \tau_{\text{OPT}}] \leq \frac{k!!}{2 \cdot 4^{\frac{k}{2}-1}(k+1)!!} \tag{16}$$

*Proof.* Let $\omega_{xy}$ denote the Hamilton Path formed by edge $e_{xy}$ and vertices $z_i \in N_c(e_{xy})$. Then, the following inequality holds:

$$\mathbb{P}[e_{xy} \in \tau_{OPT}] \leq \mathbb{P}[\text{edges in } \omega_{xy} \text{ do not cross}]. \tag{17}$$

Next, We will analyze the probability that no edges in $\omega_{xy}$ cross and obtain an upper bound. We prove this theorem by induction.

When $k = 2$, there are $z_1, z_2 \in N_c(e_{xy})$ evenly distributed on both sides of $e_{xy}$. At this time, there are six possible cases for $\omega_{xy}$: $(z_1, x, y, z_2)$, $(z_2, x, y, z_1)$, $(z_1, z_2, x, y)$, $(z_1, z_2, y, x)$, $(x, z_1, z_2, y)$, $(y, z_1, z_2, x)$. Among them, $(z_1, x, y, z_2)$ and $(z_2, x, y, z_1)$ have no edge intersection, so

$$\mathbb{P}[e_{xy} \in \tau_{OPT}] \leq \mathbb{P}[\text{edges in } \omega_{xy} \text{ do not cross}] = \frac{1}{3} = \frac{2!!}{2 \cdot 4^0 3!!} \tag{18}$$

Now, we assume that when $K_p(e_{xy}) = k$, the following equation holds:

$$\mathbb{P}[\text{edges in } \omega_{xy} \text{ do not cross}] = \frac{k!!}{4^{\frac{k}{2}-1}(k+1)!!}. \tag{19}$$

It means that among all $(k + 1)!$ combinations of $\omega_{xy}$, there are $\frac{(k!!)^2}{4^{\frac{k}{2}-1}}$ situations where the edges of $\omega_{xy}$ do not cross. When $K_p(e_{xy}) = k + 2$, first, there are totally $(k + 3)!$ combinations of $\omega_{xy}$. Next, we consider the non-intersection structure based on case $k$. For every $\frac{(k!!)^2}{4^{\frac{k}{2}-1}}$ situations

where the edges of $\omega_{xy}$ do not cross, consider one side of $e_{xy}$ first. If adding a point still leaves no intersection, there are $\frac{k}{2} + 1$ possible addition positions. Considering both sides symmetrically, there are $(\frac{k}{2} + 1)^2 = \frac{(k+2)^2}{4}$ non-intersection cases. Therefore, it can be obtained that

$$\mathbb{P}[\text{edges in } \omega_{xy} \text{ do not cross}] = \frac{\frac{(k!!)^2}{4^{k/2-1}} \cdot \frac{(k+2)^2}{4}}{(k+3)!} = \frac{(k+2)!!}{4^{(k+2)/2-1}((k+2)+1)!!} \tag{20}$$

Hence,

$$\mathbb{P}[e_{xy} \in \tau_{OPT}] \leq \frac{k!!}{4^{\frac{k}{2}-1}(k+1)!!}. \tag{21}$$

$\square$

## H  Connection between Purity Law and k-Nearest Prior

The k-nearest prior in routing problems refers to the observation that in optimal solutions, the next city visited is frequently among the $k$ nearest neighbors of the current city, where $k$ typically a small value. Purity Law and the k-nearest prior are both local phenomena that are commonly observed in optimal solutions of routing problems. However, the Purity Law can reflect more topological information.

First, the k-nearest neighbor with smaller values of that exhibit good structural quality tend to have lower purity orders for the edges they form. Second, the purity order reflects local structural information that the k-nearest prior can not capture. For example, in Fig. 1, both $x_1$ and $x_2$ are the 5-NN of $z$, and they are equivalent under the k-NN metric. However, under the purity order metric, they are distinguished. The neighbor $x_1$ with a lower purity order is a more ideal candidate for connecting to $z$ compared to $x_2$, due to its more coherent local structure. Thus, Purity Law not only contains distance information but also incorporates the structural information of the node distribution around the edges. This richer information carried by Purity Law allows it to contribute more effectively to generalization.

## I  Proof of the supermodularity of Purity Availability $\phi$

**Proposition I.1.** *The set function $\phi : 2^X \rightarrow \mathbb{R}$ defined on the subsets of the finite set $X$ is supermodular, that is, for any subset $A \subseteq B \subseteq X$ and any $x \in X \setminus B$, the following inequality holds:*

$$\phi(A \cup \{x\}) - \phi(A) \leq \phi(B \cup \{x\}) - \phi(B).$$

*Proof.* Given subsets $U \subseteq X$ and any $v_1, v_2 \in X \setminus U$, we prove the equivalent definition of supermodular functions:

$$\phi(U \cup \{v_1\}) + \phi(U \cup \{v_2\}) \leq \phi(U \cup \{v_1, v_2\}) + \phi(U). \tag{22}$$

First, we have

$$\phi(U) = \frac{\sum\limits_{x_i \in U} \min\limits_{\substack{x_j \in U \\ j \neq i}} K_p(x_i, x_j)}{|U|}.$$

$$\phi(U \cup \{v_1\}) = \frac{\sum\limits_{x_i \in U} \min\limits_{\substack{x_j \in U \cup \{v_1\} \\ j \neq i}} K_p(x_i, x_j) + \min\limits_{x_j \in U} K_p(v_1, x_j)}{|U| + 1}$$

$$\phi(U \cup \{v_2\}) = \frac{\sum\limits_{x_i \in U} \min\limits_{\substack{x_j \in U \cup \{v_2\} \\ j \neq i}} K_p(x_i, x_j) + \min\limits_{x_j \in U} K_p(v_2, x_j)}{|U| + 1}$$

$$\phi(U \cup \{v_1, v_2\}) = \frac{\sum\limits_{x_i \in U} \min\limits_{\substack{x_j \in U \cup \{v_1, v_2\} \\ j \neq i}} K_p(x_i, x_j) + \min\limits_{x_j \in U \cup \{v_2\}} K_p(v_1, x_j) + \min\limits_{x_j \in U \cup \{v_1\}} K_p(v_2, x_j)}{|U| + 2}.$$

By the following relations, we partition $U$ into three parts, denoted as $U_0, U_1, U_2$, respectively.

$$U_0 = \{x_i \mid \arg\min_y K_p(x_i, y) \in U\}$$

$$U_1 = \{x_i \mid \arg\min_y K_p(x_i, y) = v_1\}$$

$$U_2 = \{x_i \mid \arg\min_y K_p(x_i, y) = v_2\}$$

Then, we prove that the following expression is non-positive from three parts:

$$(\phi(U \cup \{v_1\}) + \phi(U \cup \{v_2\})) - (\phi(U \cup \{v_1, v_2\}) + \phi(U)).$$

**Part 1**

$$\sum_{x_i \in U_0} \min_{x_j \in U} K_p(x_i, x_j) \left[ \frac{2}{|U| + 1} - \left( \frac{1}{|U|} + \frac{1}{|U| + 2} \right) \right]$$

$$\leq \sum_{x_i \in U_0} \min_{x_j \in U} K_p(x_i, x_j) \left( \frac{-1}{|U|(|U| + 1)(|U| + 2)} \right) \leq 0$$

**Part 2**

$$\sum_{x_i \in U_1} \left( \left( \frac{\min_{x_j \in U} K_p(x_i, x_j)}{|U| + 1} - \frac{\min_{x_j \in U} K_p(x_i, x_j)}{|U|} \right) + \left( \frac{K_p(x_i, v_1)}{|U| + 1} - \frac{K_p(x_i, v_1)}{|U| + 2} \right) \right)$$

$$+ \left( \frac{\min_{x_j \in U} K_p(x_j, v_1)}{|U| + 1} - \frac{\min_{x_j \in U \cup \{v_2\}} K_p(x_j, v_1)}{|U| + 2} \right)$$

$$= \sum_{x_i \in U_1} \left( -\frac{\min_{x_j \in U} K_p(x_i, x_j)}{|U|(|U| + 1)} + \frac{K_p(x_i, v_1)}{(|U| + 1)(|U| + 2)} \right) + \frac{\min_{x_j \in U} K_p(x_j, v_1)}{(|U| + 1)(|U| + 2)}$$

$$\leq \sum_{x_i \in U_1} \left( -\frac{K_p(x_i, v_1) + 1}{|U|(|U| + 1)} + \frac{K_p(x_i, v_1)}{(|U| + 1)(|U| + 2)} \right) + \frac{\min_{x_j \in U} K_p(x_j, v_1)}{(|U| + 1)(|U| + 2)}$$

$$= \frac{1}{|U|(|U| + 1)(|U| + 2)} \left( -2 \sum_{x_j \in U_1} K_p(x_i, v_1) + |U| \min_{x_j \in U} K_p(x_j, v_1) - |U_1|(|U| + 2) \right)$$

$$\leq \frac{1}{|U|(|U| + 1)(|U| + 2)} \left( -2|U_1| \min_{x_j \in U} K_p(x_j, v_1) + |U| \min_{x_j \in U} K_p(x_j, v_1) - |U_1|(|U| + 2) \right)$$

$$\leq \frac{1}{|U|(|U| + 1)(|U| + 2)} \left( -2|U_1|^2 - 2|U_1| \right) \leq 0$$

**Part 3**

$$\sum_{x_i \in U_2} \left( \left( \frac{\min_{x_j \in U} K_p(x_i, x_j)}{|U| + 1} - \frac{\min_{x_j \in U} K_p(x_i, x_j)}{|U|} \right) + \left( \frac{K_p(x_i, v_2)}{|U| + 1} - \frac{K_p(x_i, v_2)}{|U| + 2} \right) \right)$$

$$+ \left( \frac{\min_{x_j \in U} K_p(x_j, v_2)}{|U| + 1} - \frac{\min_{x_j \in U \cup \{v_1\}} K_p(x_j, v_2)}{|U| + 2} \right)$$

$$= \sum_{x_i \in U_2} \left( -\frac{\min_{x_j \in U} K_p(x_i, x_j)}{|U|(|U| + 1)} + \frac{K_p(x_i, v_2)}{(|U| + 1)(|U| + 2)} \right) + \frac{\min_{x_j \in U} K_p(x_j, v_2)}{(|U| + 1)(|U| + 2)}$$

$$\leq \sum_{x_i \in U_2} \left( -\frac{K_p(x_i, v_2) + 1}{|U|(|U| + 1)} + \frac{K_p(x_i, v_2)}{(|U| + 1)(|U| + 2)} \right) + \frac{\min_{x_j \in U} K_p(x_j, v_2)}{(|U| + 1)(|U| + 2)}$$

$$= \frac{1}{|U|(|U| + 1)(|U| + 2)} \left( -2 \sum_{x_j \in U_2} K_p(x_i, v_2) + |U| \min_{x_j \in U} K_p(x_j, v_2) - |U_2|(|U| + 2) \right)$$

$$\leq \frac{1}{|U|(|U| + 1)(|U| + 2)} \left( -2|U_2| \min_{x_j \in U} K_p(x_j, v_2) + |U| \min_{x_j \in U} K_p(x_j, v_2) - |U_2|(|U| + 2) \right)$$

$$\leq \frac{1}{|U|(|U| + 1)(|U| + 2)} \left( -2|U_2|^2 - 2|U_2| \right) \leq 0$$

The gain consists of three parts, each of which is non-positive. Therefore, the overall gain is non-positive, and the supermodular property is thus proven.

$\square$

## J  The Whole PUPO Algorithm

The whole algorithm of PUPO is presented in Algorithm 1. PUPO is established upon the REIN-FORCE with baseline, incorporating Purity Law information into the modified policy gradient. At different stages of solution construction, PUPO introduces discounted cost information that reflects the purity potential of the partial solution and the unvisited vertex set, which encourages the model to take actions with lower purity orders at each step during the training process. This approach helps the model to learn consistent patterns with the Purity Law prior, an information that is independent of specific instances, thereby enhancing its generalization ability. Notably, PUPO can be easily integrated with arbitrary existing constructive neural solvers, without any alterations to network architecture.

## K  Theoretical analysis on optimization error of PUPO

**Theorem K.1.** *Assume that for all $\theta$, $V^{p_\theta}$ is $\beta$-smooth and bounded $V_\star$. Suppose the variance is bounded as follows: $\mathbb{E}[||\widehat{\nabla V^{p_\theta}} - \nabla V^{p_\theta}||^2] \leq \sigma^2$. For $t \leq \beta(V^* - V^{(0)})/\sigma^2$, suppose we use a constant stepsize of $\eta_t = 1/\beta$, and thereafter, we use $\eta_t = \sqrt{2/(\beta T)}$. For all $T$, we can obtain:*

$$\min_{t \leq T} \mathbb{E}[||\nabla V^{(t)}||^2] \leq \frac{2\beta(V^* - V^{(0)})}{T} + \sqrt{\frac{2\sigma^2}{T}}, \tag{23}$$

*where* $\nabla V^{p_\theta} = \mathbb{E}_{p_\theta(\tau | \mathcal{X})} \left[ (L(\tau) - b(\mathcal{X})) \sum_{t=2}^{N} W(\mathcal{U}_{t-1}, \tau_t) \nabla \log p_\theta(\tau_t | \tau_{1:t-1}, \mathcal{X}) \right]$, $\widehat{\nabla V^{p_\theta}} = \sum_{t=2}^{N} (L(\tau) - b(\mathcal{X})) W(\mathcal{U}_{t-1}, \tau_t) \nabla \log p_\theta(\tau_t | \tau_{1:t-1}, \mathcal{X})$, *and the update rule of gradient ascent algorithm is* $\theta_{t+1} = \theta_t + \eta_t \widehat{\nabla V^{p_\theta}}$.

The proof method is similar to [1]. And these optimization error theorem confirms that our proposed algorithm can converge to stationary points.

**Algorithm 1** PUPO Training

**Input:** number of epochs $E$, steps per epoch $M$, batch size $B$, scale of training instances $N$, discount factor $\gamma$, learning rate $\delta$
Init $\theta, \theta^{BL} \leftarrow \theta$
**for** epoch = 1, ..., $E$ **do**
    **for** step = 1, ..., $M$ **do**
        $\forall i \in \{1, 2, \ldots, B\}$
        $s_i \leftarrow \text{RandomInstance}()$
        $\boldsymbol{\tau}^i \leftarrow \text{SampleRollout}(s_i, p_\theta)$
        $\boldsymbol{\tau}^{i,BL} \leftarrow \text{GreedyRollout}(s_i, p_{\theta^{BL}})$
        $U_0^i \leftarrow s_i$
        **for** time = 1, ..., $N - 1$ **do**
            $U_t^i \leftarrow U_{t-1}^i \setminus \tau_t^i$
            $\phi(U_t^i) \leftarrow \sum\limits_{a \in U_t^i} \min\limits_{\substack{b \in U_t^i \\ b \neq a}} K_p(a, b, s_i)$

            $C(U_t^i, \tau_{t+1}^i) \leftarrow K_p(\tau_t^i, \tau_{t+1}^i, s_i) + \dfrac{\phi(U_{t+1}^i)}{|U_{t+1}^i|} - \dfrac{\phi(U_t^i)}{|U_t^i|}$

            $W_{t+1}^i \leftarrow 1 + \sum\limits_{j=t}^{N} \gamma^{j-t} C(U_j^i, \tau_{j+1}^i)$

        **end for**
        $\nabla\hat{\mathcal{L}} \leftarrow \dfrac{1}{B} \sum\limits_{i=1}^{B} \left( L(\boldsymbol{\tau}^i) - L(\boldsymbol{\tau}^{i,BL}) \right) \cdot \left( \sum\limits_{t=2}^{N} W_t^i \nabla_\theta \log p_\theta(\tau_t^i | \tau_{1:t-1}^i, s_i) \right)$
        $\theta \leftarrow \theta + \delta \nabla\hat{\mathcal{L}}$
    **end for**
    $\theta^{BL} \leftarrow \text{UpdateBaseline}(\theta, \theta^{BL})$
**end for**

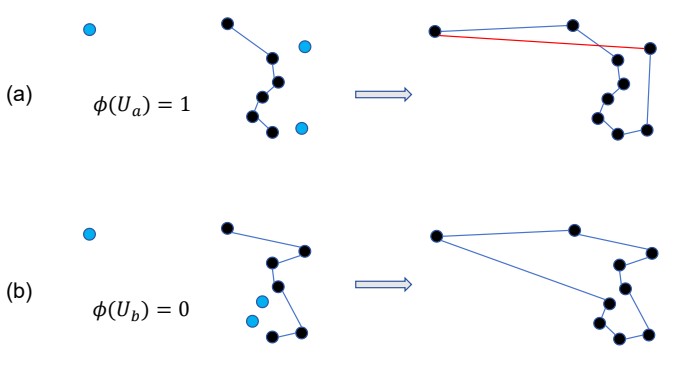

(a)    $\phi(U_a) = 1$

(b)    $\phi(U_b) = 0$

● : the vertex in unvisited vertex set      ● : the vertex in current partial tour

Figure 5: Example of two purity potential.

## L    Further Discussion on Derivation Process and Explanation of PUPO

In this section, We will discuss the derivation process and explanation of PUPO in more detail. For convenience, $W_{t+1}$ denotes the shorthand for $W(\mathcal{U}_t, \tau_{t+1})$.

**Enhancing the purity potential of the unvisited vertex set at each step is essential for learning a policy with strong generalization capability.** For intuitive understanding, we plot a specific example shown in Fig. 5. In this figure, policy (a) selects myopic local optima in the earlier decision stages. In the later stages, constrained by the requirement to select vertices only from the unvisited vertex set, it is forced to connect some "bad" edges with higher purity orders (see red edges). In contrast, strategy (b) constructs a tour with a higher overall purity level. For solving instances of the same

type during training, the tour lengths obtained by strategies (a) and (b) may not differ significantly. However, when generalized to other instances, myopic strategies like (a), which result in poor purity in the later stages, are more likely to lead to tours of lower quality.

**Inspired by supermodularity, we design purity weights to assess the potential purity of states at each time step for future decision under different policies.** We constructed a Purity Availability metric with supermodular properties to measure the purity potential of the unvisited vertex set. On one hand, since Purity Availability represents the average minimum available purity order, higher values indicate that the state will not be able to generate highly pure structures in the future. For example, in the figure above, the Purity Availability of strategies (a) and (b) at the current state are 1 and 0, respectively. It means that policy (b) possess ability to construct purer structures in the future. On the other hand, constrained by the necessity to select nodes only from the unvisited vertex set, later decision stages often lead to states that are unfavorable for generalization and difficult to distinguish under tour length metrics. Due to supermodularity, the marginal benefit of Purity Availability, $\phi(\mathcal{U}_t) - \phi(\mathcal{U}_{t-1})$, is greater at later stages, making the purity measure in the later decision phases more significant. Inspired by supermodularity, we designed the purity cost based on the marginal benefit of state and the purity order of the actions. Furthermore, we designed purity weights based on the concept of future discounting to comprehensively assess the future purity potential of intermediate states during policy transitions.

**Through analysis of upper and lower bounds and experimental analysis, PUPO encourages the exploration of higher-quality states under the purity measure.** First, we present a pair of upper and lower bounds for Eq. (12). Since $W_t \geq 1$, the following inequality holds:

$$E_{p_\theta(\boldsymbol{\tau}|\mathcal{X})}\left[L(\boldsymbol{\tau})\sum_{t=2}^{N}\nabla\log p_\theta(\tau_t|\tau_{1:t-1},\mathcal{X})\right]$$

$$\leq E_{p_\theta(\boldsymbol{\tau}|\mathcal{X})}\left[L(\boldsymbol{\tau})\sum_{t=2}^{N}W_t\nabla\log p_\theta(\tau_t|\tau_{1:t-1},\mathcal{X})\right]$$

$$\leq E_{p_\theta(\boldsymbol{\tau}|\mathcal{X})}\left[L(\boldsymbol{\tau})\left(\sum_{t=2}^{N}W_t\right)\sum_{t=2}^{N}\nabla\log p_\theta(\tau_t|\tau_{1:t-1},\mathcal{X})\right].$$

The left-hand side of the inequality represents the original policy gradient with tour length as the reward, while the right-hand side represents the policy gradient with $\left(\sum_{t=2}^{N}W_t\right)L(\boldsymbol{\tau})$ as the reward.

In this reward, the purity weights act as multiplicative factors on the tour length. It means that the overall purity of the tour is also incorporated into the reward information. The policy gradient of PUPO lies between the two, applying each $W_t$ to the logarithm of action probabilities at each time step. This more subtle approach encourages the exploration of strategies that can transition to states with higher quality under the purity measure.

## M   Numerical Result on Two Classical Solvers

We also conduct experiments on two classical neural solvers POMO and PF. Table 9 show the results on random dataset of TSP. It can be observed that PUPO training enhances the performance of four classical models across nearly all instance types. In particular, AM-100 and PF-100 demonstrate superior performance across all distributions and scales compared to those trained with the vanilla. Table 10 also shows the results on TSPLIB, and Table 11 shows the result on random dataset of CVRP. It can be observed that for CVRP, PUPO training enhances the performance more significantly. Although PUPO also enhanced the generalizability of AM and PF, their global perception and decision modules remain vulnerable, resulting in larger performance gaps as the scale increases. This observation abstracts us to focus on developing network architectures that incorporate Purity Law in future work.

Table 9: The experimental results of average gap (%) on the randomly generated dataset with different distributions and scales of TSP after training two classical model using both the vanilla and PUPO methods, where bold formatting represents superior results.

| Instances | POMO-50 | | POMO-100 | | PF-50 | | PF-100 | |
|---|---|---|---|---|---|---|---|---|
| | Vanilla | PUPO | Vanilla | PUPO | Vanilla | PUPO | Vanilla | PUPO |
| U-100 | **5.21** | 5.35 | 5.07 | **4.61** | **4.16** | 4.31 | 3.64 | **3.39** |
| U-1000 | 35.51 | **34.01** | 31.23 | **30.16** | 41.00 | **39.10** | 30.57 | **29.93** |
| U-5000 | 67.99 | **65.12** | 80.64 | **68.42** | 113.55 | **104.09** | 80.59 | **77.53** |
| C-100 | **7.75** | 8.03 | 7.66 | **6.07** | **7.67** | 7.71 | 7.24 | **6.90** |
| C-1000 | 41.14 | **39.82** | 37.99 | **33.84** | 51.04 | **49.58** | 47.50 | **44.11** |
| C-5000 | 77.78 | **75.93** | 76.37 | **70.03** | 137.51 | **131.61** | 137.15 | **121.79** |
| E-100 | **5.01** | 5.18 | 5.04 | **4.71** | **4.98** | 5.12 | 4.74 | **4.65** |
| E-1000 | 35.57 | **35.07** | 32.97 | **30.20** | 46.04 | **45.34** | 41.60 | **37.73** |
| E-5000 | 71.95 | **66.66** | 74.55 | **66.33** | 133.99 | **129.02** | 120.96 | **110.16** |
| I-100 | **5.02** | 5.17 | 5.12 | **4.64** | **4.71** | 4.91 | 4.49 | **4.35** |
| I-1000 | 37.38 | **37.12** | 32.44 | **30.94** | 46.54 | **44.83** | 36.11 | **34.47** |
| I-5000 | 78.57 | **74.05** | 72.87 | **66.58** | 151.37 | **138.71** | 120.67 | **103.74** |

Table 10: Performance of average gap (%) on TSPLIB after training two classical model using both the vanilla and PUPO methods.

| | POMO-50 | | POMO-100 | | PF-50 | | PF-100 | |
|---|---|---|---|---|---|---|---|---|
| | Vanilla | PUPO | Vanilla | PUPO | Vanilla | PUPO | Vanilla | PUPO |
| 1~100 | 5.82 | 5.17 | 4.98 | 4.48 | 7.42 | 8.37 | 7.82 | 7.16 |
| 101~1000 | 17.39 | 16.14 | 14.12 | 12.17 | 25.96 | 24.77 | 23.37 | 22.68 |
| 1001~5000 | 48.41 | 46.48 | 43.33 | 40.46 | 75.57 | 72.58 | 66.05 | 60.86 |

Table 11: The experimental results of average gap (%) on the randomly generated dataset with different distributions and scales of CVRP after training POMO model using both the vanilla and PUPO methods, where bold formatting represents superior results.

| Instance | POMO-50 | | POMO-100 | |
|---|---|---|---|---|
| | Vanilla | PUPO | Vanilla | PUPO |
| U-50 | **4.6** | 5.704 | **6.146** | 6.769 |
| U-500 | 16.379 | **14.311** | 14.843 | **13.383** |
| U-5000 | 22.846 | **16.122** | 20.599 | **18.067** |
| C-50 | **5.204** | 6.346 | 6.388 | **6.384** |
| C-500 | 14.997 | **13.21** | 12.837 | **12.279** |
| C-5000 | 15.803 | **11.694** | **13.172** | 13.376 |
| E-50 | **4.624** | 5.851 | **6.042** | 6.704 |
| E-500 | 16.227 | **14.622** | 14.249 | **13.525** |
| E-5000 | 22.761 | **17.148** | 21.638 | **17.673** |
| I-50 | **4.57** | 5.794 | **6.117** | 6.911 |
| I-500 | 15.744 | **14.239** | 14.033 | **13.16** |
| I-5000 | 19.703 | **14.64** | 16.513 | **16.214** |

# N  Detailed Description of Dataset in Comparative Experiments

To ensure the adequacy of the experiments, we validate the model's performance on two categories of datasets. The randomly generated dataset used in this paper is the same as that in INViT [10], which is widely adopted to testify existing DRL approach. For TSP, it contains 16 subsets and corresponding (near-)optimal solutions for TSP, including 4 distributions (uniform, clustered, explosion, and implosion, denoted as U, C, E, I) and 4 scales (100, 1000, 5000 and 10000). For CVRP, it contains 12 subsets and corresponding (near-)optimal solutions for CVRP, including 4 distributions (uniform, clustered, explosion, and implosion, denoted as U, C, E, I) and 3 scales (50, 500 and 5000).

The real-world dataset we used is TSPLIB and CVRPLIB. TSPLIB is a well-known TSP library [39] that contains 100 instances with various nodes distributions and their optimal solutions. These instances come from practical applications with scale ranging from 14 to 85,900. In our experiment, we consider all instances with no more than 10000 nodes. For CVRP, we include all instances in CVRPLIB Set-X [40], containing 100 instances varying in scale from 100 to 1000.

## O    Detailed of the Learning Rates of Models

Specifically, for TSP, the learning rates are set to 0.0001, 0.00015, 0.0001, 0.00017, 0.0001, 0.00012, 0.00011, 0.00012 for POMO-50, POMO-100, PF-50, PF-100, ELG-50, ELG-100, INViT-50, and INViT-100, respectively. For CVRP, the learning rates are set to 0.0001, 0.00006, 0.0001, 0.0001, 0.00005, 0.0001 for POMO-50, POMO-100, ELG-50, ELG-100, INViT-50, and INViT-100.

## P    Detailed Experimental Result

Table 12 shows the training time per epoch with different methods. We can see that the training time of PUPO does not increase significantly owing to the tensorizable computation.

Table 12: The numerical results of training time (minutes) per epoch during different training.

|  | POMO-50 | POMO-100 | INViT-50 | INViT-100 |
|---|---|---|---|---|
| Vanilla | 3.05 | 4.49 | 7.72 | 15.75 |
| PUPO | 5.95 | 14.20 | 11.598 | 21.40 |

The experimental results of the specific tour length is presented in Table 13. Table 14 illustrates the solving time of Vanilla-trained and PUPO-trained models on randomly generated dataset. Following the table, it can be observed that there is almost no difference between Vanilla and PUPO. The numerical results of purity metrics for all models are presented in Table 15. Following the table, PUPO-trained models possess the ability to generate solutions with more outstanding purity. Detailed results of ELG and INViT are provided in Table 16 to Table 27.

Table 13: The length of tours generated from each model after different training on randomly generated dataset of TSP.

|  |  | POMO-50 | | POMO-100 | | PF-50 | | PF-100 | | INViT-50 | | INViT-100 | |
|---|---|---|---|---|---|---|---|---|---|---|---|---|---|
|  |  | Vanilla | PUPO | Vanilla | PUPO | Vanilla | PUPO | Vanilla | PUPO | Vanilla | PUPO | Vanilla | PUPO |
| Uniform | 100 | 8.28 | 8.29 | 8.27 | 8.23 | 8.19 | 8.21 | 8.15 | 8.13 | 8.03 | 8.04 | 8.07 | 8.05 |
|  | 1000 | 31.47 | 31.12 | 30.47 | 30.22 | 32.74 | 32.30 | 30.32 | 30.17 | 24.91 | 24.88 | 24.94 | 24.67 |
|  | 5000 | 85.75 | 84.28 | 92.20 | 85.97 | 109.00 | 104.18 | 92.18 | 90.62 | 55.85 | 55.69 | 55.84 | 55.15 |
|  | 10000 | - | - | - | - | - | - | - | - | 79.07 | 78.88 | 78.79 | 77.87 |
| Clustered | 100 | 5.72 | 5.73 | 5.71 | 5.63 | 5.71 | 5.71 | 5.68 | 5.67 | 5.46 | 5.46 | 5.50 | 5.48 |
|  | 1000 | 19.82 | 19.65 | 19.39 | 18.80 | 21.26 | 21.05 | 20.78 | 20.30 | 15.25 | 15.19 | 15.24 | 15.07 |
|  | 5000 | 53.06 | 52.50 | 52.64 | 50.77 | 70.86 | 69.10 | 70.71 | 66.13 | 32.79 | 32.66 | 32.73 | 32.39 |
|  | 10000 | - | - | - | - | - | - | - | - | 44.88 | 44.58 | 44.72 | 44.17 |
| Explosion | 100 | 6.86 | 6.87 | 6.86 | 6.83 | 6.86 | 6.87 | 6.84 | 6.83 | 6.67 | 6.66 | 6.70 | 6.68 |
|  | 1000 | 21.90 | 21.84 | 21.47 | 21.03 | 23.50 | 23.36 | 22.72 | 22.10 | 17.67 | 17.60 | 17.67 | 17.53 |
|  | 5000 | 57.30 | 55.81 | 58.14 | 55.62 | 77.35 | 75.30 | 72.57 | 69.02 | 37.52 | 37.28 | 37.23 | 36.88 |
|  | 10000 | - | - | - | - | - | - | - | - | 43.28 | 43.05 | 43.18 | 42.66 |
| Implosion | 100 | 7.48 | 7.49 | 7.48 | 7.45 | 7.45 | 7.47 | 7.43 | 7.42 | 7.29 | 7.29 | 7.32 | 7.30 |
|  | 1000 | 27.61 | 27.57 | 26.63 | 26.33 | 29.39 | 29.04 | 27.28 | 26.95 | 21.64 | 21.63 | 21.69 | 21.51 |
|  | 5000 | 72.59 | 71.24 | 71.28 | 68.46 | 100.18 | 95.14 | 86.94 | 80.27 | 44.96 | 44.83 | 45.05 | 44.78 |
|  | 10000 | - | - | - | - | - | - | - | - | 72.24 | 71.79 | 72.17 | 71.18 |

Table 14: The solving time for each model with different training methods on randomly generated dataset of TSP.

| | POMO-50 | | POMO-100 | | PF-50 | | PF-100 | | INViT-50 | | INViT-100 | |
|---|---|---|---|---|---|---|---|---|---|---|---|---|
| | Vanilla | PUPO | Vanilla | PUPO | Vanilla | PUPO | Vanilla | PUPO | Vanilla | PUPO | Vanilla | PUPO |
| U-100 | 0.20 | 0.22 | 0.21 | 0.22 | 2.02 | 1.97 | 1.00 | 1.02 | 0.85 | 0.80 | 1.07 | 0.91 |
| U-1000 | 2.35 | 2.34 | 2.41 | 2.40 | 23.67 | 24.08 | 13.57 | 14.18 | 13.24 | 12.88 | 17.83 | 17.34 |
| U-5000 | 19.68 | 19.10 | 20.00 | 19.57 | 246.65 | 246.12 | 205.14 | 204.15 | 91.20 | 89.44 | 120.07 | 119.87 |
| U-10000 | - | - | - | - | - | - | - | - | 224.99 | 223.69 | 288.44 | 287.11 |
| C-100 | 0.21 | 0.21 | 0.21 | 0.21 | 2.08 | 2.13 | 0.97 | 0.98 | 0.84 | 0.80 | 1.04 | 1.17 |
| C-1000 | 2.35 | 2.35 | 2.44 | 2.43 | 25.28 | 26.11 | 14.05 | 15.28 | 13.00 | 12.77 | 17.57 | 16.56 |
| C-5000 | 19.16 | 19.38 | 19.99 | 19.35 | 275.68 | 274.49 | 212.36 | 211.68 | 89.48 | 88.23 | 118.63 | 117.31 |
| C-10000 | - | - | - | - | - | - | - | - | 222.47 | 222.32 | 288.67 | 288.90 |
| E-100 | 0.21 | 0.21 | 0.20 | 0.22 | 2.18 | 2.17 | 0.97 | 1.09 | 0.82 | 0.83 | 1.05 | 0.94 |
| E-1000 | 2.38 | 2.36 | 2.46 | 2.40 | 25.44 | 24.53 | 13.94 | 13.82 | 12.94 | 13.05 | 17.70 | 17.98 |
| E-5000 | 19.30 | 18.67 | 19.98 | 19.57 | 411.08 | 411.22 | 213.01 | 212.39 | 89.58 | 89.93 | 118.93 | 118.34 |
| E-10000 | - | - | - | - | - | - | - | - | 223.24 | 223.03 | 288.86 | 288.32 |
| I-100 | 0.21 | 0.20 | 0.21 | 0.21 | 3.02 | 2.89 | 1.00 | 0.97 | 0.83 | 0.79 | 1.05 | 1.01 |
| I-1000 | 2.37 | 2.45 | 2.45 | 2.45 | 20.84 | 21.01 | 14.08 | 13.99 | 12.91 | 12.73 | 17.60 | 17.31 |
| I-5000 | 19.95 | 19.28 | 19.84 | 19.30 | 445.75 | 444.87 | 186.14 | 186.05 | 89.92 | 88.30 | 118.87 | 118.69 |
| I-10000 | - | - | - | - | - | - | - | - | 222.56 | 221.72 | 289.39 | 290.99 |

Table 15: The numerical results of purity evaluation for three models on TSP after different training.

POMO-50

| | Prop-0 (%) | Vanilla APO (all) | APO (non-0) | Prop-0 (%) | PUPO APO (all) | APO (non-0) |
|---|---|---|---|---|---|---|
| U-100 | 75.65% | 0.15 | 1.11 | 75.29% | 0.15 | 1.11 |
| U-1000 | 62.51% | 0.64 | 1.98 | 64.32% | 0.62 | 1.92 |
| U-5000 | 24.91% | 1.36 | 2.97 | 26.39% | 1.33 | 3.05 |
| U-10000 | - | - | - | - | - | - |
| C-100 | 70.66% | 0.23 | 1.26 | 70.13% | 0.24 | 1.28 |
| C-1000 | 59.13% | 0.85 | 2.30 | 58.29% | 0.80 | 2.19 |
| C-5000 | 23.65% | 1.93 | 3.88 | 23% | 1.64 | 3.33 |
| C-10000 | - | - | - | - | - | - |
| E-100 | 73.72% | 0.17 | 1.15 | 74.35% | 0.17 | 1.14 |
| E-1000 | 62.92% | 0.69 | 2.03 | 61.35% | 0.67 | 2.03 |
| E-5000 | 25.09% | 1.53 | 3.27 | 23.99% | 1.41 | 3.06 |
| E-10000 | - | - | - | - | - | - |
| I-100 | 72.97% | 0.19 | 1.20 | 73.50% | 0.19 | 1.19 |
| I-1000 | 61.52% | 0.77 | 2.22 | 60.27% | 0.77 | 2.20 |
| I-5000 | 24.05% | 1.93 | 3.83 | 23.70% | 1.69 | 3.41 |
| I-10000 | - | - | - | - | - | - |

POMO-100

| | Prop-0 (%) | Vanilla APO (all) | APO (non-0) | Prop-0 (%) | PUPO APO (all) | APO (non-0) |
|---|---|---|---|---|---|---|
| U-100 | 76.81% | 0.15 | 1.12 | 79.48% | 0.14 | 1.17 |
| U-1000 | 65.67% | 0.52 | 1.75 | 67.70% | 0.52 | 1.76 |
| U-5000 | 24.54% | 1.50 | 3.03 | 26.64% | 1.23 | 2.74 |
| U-10000 | - | - | - | - | - | - |
| C-100 | 72.34% | 0.24 | 1.30 | 75.35% | 0.20 | 1.28 |
| C-1000 | 61.18% | 0.77 | 2.17 | 62.52% | 0.66 | 1.97 |
| C-5000 | 23.58% | 1.64 | 3.31 | 24.76% | 1.48 | 3.05 |
| C-10000 | - | - | - | - | - | - |
| E-100 | 75.24% | 0.17 | 1.17 | 77.77% | 0.17 | 1.22 |
| E-1000 | 63.32% | 0.62 | 1.89 | 65.58% | 0.57 | 1.84 |
| E-5000 | 24.20% | 1.55 | 3.13 | 26.04% | 1.30 | 2.82 |
| E-10000 | - | - | - | - | - | - |
| I-100 | 74.80% | 0.19 | 1.23 | 77.27% | 0.18 | 1.26 |
| I-1000 | 64.14% | 0.65 | 2.01 | 65.01% | 0.62 | 1.95 |
| I-5000 | 24.12% | 1.65 | 3.33 | 25.30% | 1.46 | 3.07 |
| I-10000 | - | - | - | - | - | - |

PF-50

| | Prop-0 (%) | Vanilla APO (all) | APO (non-0) | Prop-0 (%) | PUPO APO (all) | APO (non-0) |
|---|---|---|---|---|---|---|
| U-100 | 77.46% | 0.14 | 1.09 | 77.57% | 0.15 | 1.10 |
| U-1000 | 59.68% | 0.74 | 2.00 | 60.73% | 0.70 | 1.94 |
| U-5000 | 18.66% | 2.36 | 3.92 | 20.04% | 2.14 | 3.65 |
| U-10000 | - | - | - | - | - | - |
| C-100 | 71.41% | 0.24 | 1.27 | 71.53% | 0.25 | 1.29 |
| C-1000 | 52.39% | 1.21 | 2.67 | 52.53% | 1.17 | 2.60 |
| C-5000 | 15.08% | 4.52 | 6.54 | 15.42% | 4.06 | 5.95 |
| C-10000 | - | - | - | - | - | - |
| E-100 | 74.92% | 0.19 | 1.16 | 74.82% | 0.18 | 1.17 |
| E-1000 | 55.35% | 0.98 | 2.33 | 55.95% | 0.98 | 2.34 |
| E-5000 | 16.05% | 3.52 | 5.18 | 16.64% | 3.39 | 5.10 |
| E-10000 | - | - | - | - | - | - |
| I-100 | 73.76% | 0.24 | 1.30 | 73.68% | 0.24 | 1.28 |
| I-1000 | 54.17% | 1.40 | 3.09 | 55.04% | 1.42 | 3.15 |
| I-5000 | 15.18% | 8.14 | 10.81 | 15.90% | 8.46 | 11.59 |
| I-10000 | - | - | - | - | - | - |

PF-100

| | Prop-0 (%) | Vanilla APO (all) | APO (non-0) | Prop-0 (%) | PUPO APO (all) | APO (non-0) |
|---|---|---|---|---|---|---|
| U-100 | 80.07% | 0.12 | 1.08 | 80.31% | 0.12 | 1.08 |
| U-1000 | 65.66% | 0.56 | 1.78 | 66.06% | 0.54 | 1.75 |
| U-5000 | 23.09% | 1.56 | 2.94 | 23.60% | 1.49 | 2.92 |
| U-10000 | - | - | - | - | - | - |
| C-100 | 72.49% | 0.24 | 1.29 | 73.64% | 0.23 | 1.28 |
| C-1000 | 54.38% | 1.09 | 2.54 | 56.51% | 0.97 | 2.37 |
| C-5000 | 16.08% | 4.20 | 6.31 | 18% | 3.13 | 4.93 |
| C-10000 | - | - | - | - | - | - |
| E-100 | 76.63% | 0.17 | 1.17 | 77.34% | 0.16 | 1.16 |
| E-1000 | 57.53% | 0.89 | 2.21 | 59.66% | 0.81 | 2.12 |
| E-5000 | 17.60% | 3.01 | 4.62 | 18.77% | 2.73 | 4.40 |
| E-10000 | - | - | - | - | - | - |
| I-100 | 75.67% | 0.22 | 1.29 | 76.24% | 0.22 | 1.29 |
| I-1000 | 59.29% | 1.14 | 2.79 | 60.07% | 1.13 | 2.81 |
| I-5000 | 17.79% | 7.46 | 10.38 | 18.88% | 6.02 | 8.88 |
| I-10000 | - | - | - | - | - | - |

INViT-50

| | Prop-0 (%) | Vanilla APO (all) | APO (non-0) | Prop-0 (%) | PUPO APO (all) | APO (non-0) |
|---|---|---|---|---|---|---|
| U-100 | 81.52% | 0.10 | 1.05 | 0.82 | 0.10 | 1.04 |
| U-1000 | 86% | 0.16 | 1.37 | 0.86 | 0.15 | 1.33 |
| U-5000 | 43.52% | 0.18 | 1.56 | 0.44 | 0.17 | 1.05 |
| U-10000 | 87.39% | 0.18 | 1.55 | 0.88 | 0.17 | 1.50 |
| C-100 | 79.78% | 0.14 | 1.18 | 0.80 | 0.13 | 1.17 |
| C-1000 | 86.02% | 0.22 | 1.88 | 0.86 | 0.19 | 1.63 |
| C-5000 | 43.72% | 0.20 | 1.61 | 0.44 | 0.18 | 1.55 |
| C-10000 | 87.40% | 0.84 | 6.91 | 0.88 | 0.88 | 7.34 |
| E-100 | 80.56% | 0.12 | 1.11 | 0.81 | 0.12 | 1.11 |
| E-1000 | 85.42% | 0.30 | 2.34 | 0.86 | 0.28 | 2.23 |
| E-5000 | 43.54% | 0.42 | 3.47 | 0.44 | 0.48 | 4.07 |
| E-10000 | 87% | 0.95 | 7.68 | 0.87 | 1.03 | 8.38 |
| I-100 | 79.98% | 0.12 | 1.12 | 0.80 | 0.12 | 1.12 |
| I-1000 | 85.69% | 0.19 | 1.54 | 0.86 | 0.18 | 1.54 |
| I-5000 | 43.39% | 0.23 | 1.88 | 0.44 | 0.22 | 1.81 |
| I-10000 | 87.34% | 0.30 | 2.47 | 0.88 | 0.29 | 2.39 |

INViT-100

| | Prop-0 (%) | Vanilla APO (all) | APO (non-0) | Prop-0 (%) | PUPO APO (all) | APO (non-0) |
|---|---|---|---|---|---|---|
| U-100 | 80.95% | 0.11 | 1.07 | 0.82 | 0.10 | 1.05 |
| U-1000 | 85.80% | 0.17 | 1.43 | 0.87 | 0.14 | 1.33 |
| U-5000 | 43.38% | 0.20 | 1.61 | 0.44 | 0.17 | 1.48 |
| U-10000 | 87.32% | 0.20 | 1.65 | 0.88 | 0.16 | 1.47 |
| C-100 | 79.26% | 0.16 | 1.26 | 0.80 | 0.14 | 1.19 |
| C-1000 | 85.56% | 0.23 | 1.83 | 0.87 | 0.18 | 1.61 |
| C-5000 | 43.32% | 0.27 | 2.20 | 0.44 | 0.17 | 1.52 |
| C-10000 | 87.15% | 0.71 | 5.74 | 0.88 | 0.87 | 7.77 |
| E-100 | 80.09% | 0.13 | 1.15 | 0.81 | 0.12 | 1.11 |
| E-1000 | 85.09% | 0.30 | 2.33 | 0.86 | 0.26 | 2.16 |
| E-5000 | 43.40% | 0.53 | 4.33 | 0.44 | 0.43 | 3.73 |
| E-10000 | 87% | 0.97 | 7.58 | 0.88 | 0.88 | 7.57 |
| I-100 | 79.77% | 0.14 | 1.17 | 0.80 | 0.13 | 1.13 |
| I-1000 | 85.39% | 0.21 | 1.64 | 0.86 | 0.18 | 1.55 |
| I-5000 | 43.33% | 0.24 | 1.92 | 0.44 | 0.19 | 1.66 |
| I-10000 | 87.16% | 0.27 | 2.18 | 0.88 | 0.30 | 2.62 |

Table 16: Detailed results of ELG-TSP-50 [Vanilla] on random dataset.

| Instance | Mean Time (s) | Mean Length | Mean Gap (%) | Min Gap (%) | Max Gap (%) | Std |
|---|---|---|---|---|---|---|
| U-50 | 0.167 | 8.385 | 6.599 | 1.095 | 13.377 | 0.019 |
| U-500 | 1.638 | 28.051 | 20.805 | 16.915 | 25.198 | 0.017 |
| U-5000 | 21.006 | 67.121 | 31.496 | 30.52 | 32.456 | 0.009 |
| C-50 | 0.17 | 5.838 | 10.047 | 3.552 | 20.179 | 0.03 |
| C-500 | 2.027 | 17.867 | 27.142 | 19.898 | 34.527 | 0.034 |
| C-5000 | 19.721 | 41.468 | 38.888 | 35.129 | 42.994 | 0.03 |
| E-50 | 0.169 | 6.97 | 6.848 | 1.456 | 19.511 | 0.024 |
| E-500 | 2.079 | 20.064 | 24.554 | 17.191 | 32.079 | 0.036 |
| E-5000 | 19.915 | 45.129 | 35.672 | 32.112 | 42.508 | 0.045 |
| I-50 | 0.167 | 7.591 | 6.655 | 1.143 | 14.219 | 0.021 |
| I-500 | 1.96 | 24.401 | 21.303 | 17.254 | 25.035 | 0.02 |
| I-5000 | 19.832 | 53.693 | 30.403 | 28.01 | 32.592 | 0.018 |

Table 17: Detailed results of ELG-TSP-50 [PUPO] on random dataset.

| Instance | Mean Time (s) | Mean Length | Mean Gap (%) | Min Gap (%) | Max Gap (%) | Std |
|---|---|---|---|---|---|---|
| U-50 | 0.163 | 8.425 | 7.102 | 0.73 | 13.637 | 0.018 |
| U-500 | 1.667 | 27.742 | 19.474 | 15.497 | 22.476 | 0.015 |
| U-5000 | 21.34 | 64.744 | 26.84 | 24.991 | 28.767 | 0.015 |
| C-50 | 0.169 | 5.783 | 8.982 | 2.337 | 17.056 | 0.026 |
| C-500 | 2.012 | 17.461 | 24.132 | 18.047 | 29.298 | 0.024 |
| C-5000 | 19.84 | 40.475 | 35.625 | 34.174 | 38.202 | 0.016 |
| E-50 | 0.171 | 6.996 | 7.234 | 0.849 | 15.83 | 0.022 |
| E-500 | 2.139 | 19.941 | 23.767 | 18.862 | 30.275 | 0.032 |
| E-5000 | 19.73 | 42.989 | 28.882 | 26.586 | 32.44 | 0.022 |
| I-50 | 0.172 | 7.629 | 7.233 | 1.79 | 19.087 | 0.023 |
| I-500 | 2 | 24.238 | 20.544 | 16.96 | 25.203 | 0.019 |
| I-5000 | 19.781 | 52.342 | 27.644 | 25.039 | 30.885 | 0.021 |

Table 18: Detailed results of ELG-TSP-100 [Vanilla] on random dataset.

| Instance | Mean Time (s) | Mean Length | Mean Gap (%) | Min Gap (%) | Max Gap (%) | Std |
|---|---|---|---|---|---|---|
| U-50 | 0.17 | 8.327 | 5.859 | 1.414 | 10.455 | 0.017 |
| U-500 | 1.714 | 27.389 | 17.954 | 15.157 | 21.448 | 0.016 |
| U-5000 | 21.802 | 65.96 | 29.223 | 27.378 | 30.284 | 0.011 |
| C-50 | 0.18 | 5.912 | 11.495 | 3.5 | 33.416 | 0.041 |
| C-500 | 2.135 | 17.842 | 27.057 | 17.086 | 53.627 | 0.059 |
| C-5000 | 20.303 | 41.904 | 40.44 | 35.008 | 53.044 | 0.073 |
| E-50 | 0.177 | 6.991 | 7.277 | 1.466 | 32.814 | 0.037 |
| E-500 | 2.164 | 19.952 | 24.007 | 15.306 | 34.444 | 0.046 |
| E-5000 | 20.288 | 44.711 | 34.771 | 28.244 | 53.191 | 0.104 |
| I-50 | 0.181 | 7.582 | 6.639 | 1.27 | 53.637 | 0.037 |
| I-500 | 2.129 | 23.869 | 18.621 | 13.569 | 23.293 | 0.021 |
| I-5000 | 20.454 | 53.499 | 30.701 | 28.03 | 37.713 | 0.04 |

Table 19: Detailed results of ELG-TSP-100 [PUPO] on random dataset.

| Instance | Mean Time (s) | Mean Length | Mean Gap (%) | Min Gap (%) | Max Gap (%) | Std |
|---|---|---|---|---|---|---|
| U-50 | 0.169 | 8.416 | 6.394 | 1.544 | 11.726 | 0.018 |
| U-500 | 1.642 | 27.513 | 18.491 | 15.788 | 21.281 | 0.013 |
| U-5000 | 21.291 | 63.176 | 23.769 | 22.484 | 24.722 | 0.008 |
| C-50 | 0.168 | 5.803 | 9.374 | 3.302 | 19.165 | 0.027 |
| C-500 | 2.028 | 17.033 | 21.131 | 15.879 | 26.933 | 0.022 |
| C-5000 | 19.813 | 38.576 | 29.285 | 27.614 | 30.616 | 0.011 |
| E-50 | 0.17 | 6.993 | 7.188 | 1.913 | 15.156 | 0.022 |
| E-500 | 2.094 | 19.599 | 21.568 | 16.944 | 27.233 | 0.025 |
| E-5000 | 19.835 | 42.083 | 26.262 | 23.14 | 29.791 | 0.025 |
| I-50 | 0.171 | 7.657 | 7.602 | 1.684 | 16.868 | 0.023 |
| I-500 | 1.978 | 24.035 | 19.515 | 14.306 | 23.205 | 0.017 |
| I-5000 | 19.735 | 51.01 | 24.636 | 21.293 | 28.75 | 0.031 |

Table 20: Detailed results of ELG-VRP-50 [Vanilla] on random dataset.

| Instance | Mean Time (s) | Mean Length | Mean Gap (%) | Min Gap (%) | Max Gap (%) | Std |
|---|---|---|---|---|---|---|
| U-50 | 0.182 | 10.597 | 7.446 | 2.438 | 15.163 | 0.024 |
| U-500 | 1.702 | 79.532 | 15.478 | 10.783 | 20.443 | 0.027 |
| U-5000 | 41.938 | 729.196 | 22.753 | 12.2 | 31.278 | 0.069 |
| C-50 | 0.192 | 9.228 | 7.451 | 1.417 | 19.272 | 0.032 |
| C-500 | 1.788 | 64.274 | 16.468 | 10.468 | 25.644 | 0.035 |
| C-5000 | 31.429 | 681.954 | 20.448 | 14.562 | 29.403 | 0.063 |
| E-50 | 0.189 | 9.461 | 7.577 | 1.767 | 16.394 | 0.026 |
| E-500 | 1.783 | 60.506 | 15.855 | 11.473 | 26.178 | 0.032 |
| E-5000 | 31.684 | 437.246 | 27.103 | 15.508 | 48.303 | 0.126 |
| I-50 | 0.191 | 10.086 | 7.766 | 1.961 | 19.897 | 0.028 |
| I-500 | 1.797 | 73.173 | 15.762 | 10.151 | 22.652 | 0.03 |
| I-5000 | 31.711 | 700.698 | 19.427 | 15.743 | 28.109 | 0.051 |

Table 21: Detailed results of ELG-VRP-50 [PUPO] on random dataset.

| Instance | Mean Time (s) | Mean Length | Mean Gap (%) | Min Gap (%) | Max Gap (%) | Std |
|---|---|---|---|---|---|---|
| U-50 | 0.186 | 10.665 | 8.15 | 2.483 | 21.006 | 0.027 |
| U-500 | 1.694 | 76.488 | 10.968 | 8.122 | 15.112 | 0.016 |
| U-5000 | 31.128 | 647.568 | 8.312 | 7.469 | 10.66 | 0.013 |
| C-50 | 0.189 | 9.21 | 7.275 | 1.289 | 20.119 | 0.029 |
| C-500 | 1.759 | 60.866 | 10.214 | 6.542 | 15.321 | 0.02 |
| C-5000 | 30.684 | 615.111 | 8.72 | 4.76 | 16.811 | 0.048 |
| E-50 | 0.189 | 9.513 | 8.132 | 2.725 | 17.49 | 0.027 |
| E-500 | 1.748 | 58.13 | 11.117 | 8.009 | 17.598 | 0.02 |
| E-5000 | 30.737 | 373.572 | 8.516 | 6.827 | 11.048 | 0.018 |
| I-50 | 0.187 | 10.127 | 8.22 | 2.307 | 17.804 | 0.028 |
| I-500 | 1.767 | 70.051 | 10.83 | 7.665 | 14.91 | 0.016 |
| I-5000 | 30.623 | 629.346 | 7.046 | 4.803 | 9.885 | 0.021 |

Table 22: Detailed results of ELG-VRP-100 [Vanilla] on random dataset.

| Instance | Mean Time (s) | Mean Length | Mean Gap (%) | Min Gap (%) | Max Gap (%) | Std |
|---|---|---|---|---|---|---|
| U-50 | 0.195 | 10.685 | 8.315 | 1.049 | 19.544 | 0.028 |
| U-500 | 1.843 | 75.079 | 8.907 | 6.067 | 13.065 | 0.015 |
| U-5000 | 33.265 | 631.606 | 5.969 | 4.003 | 8.587 | 0.018 |
| C-50 | 0.199 | 9.22 | 7.395 | 1.519 | 21.493 | 0.032 |
| C-500 | 1.856 | 60.128 | 8.818 | 4.944 | 13.067 | 0.018 |
| C-5000 | 31.53 | 609.493 | 7.103 | 5.912 | 8.039 | 0.01 |
| E-50 | 0.2 | 9.547 | 8.566 | 2.14 | 23.925 | 0.031 |
| E-500 | 1.865 | 57.156 | 9.322 | 7.064 | 15.291 | 0.019 |
| E-5000 | 31.831 | 372.201 | 8.259 | 6.408 | 9.693 | 0.013 |
| I-50 | 0.201 | 10.144 | 8.344 | 1.643 | 20.51 | 0.031 |
| I-500 | 1.888 | 68.729 | 8.97 | 5.076 | 14.439 | 0.017 |
| I-5000 | 31.839 | 622.795 | 6.013 | 5.351 | 6.99 | 0.007 |

Table 23: Detailed results of ELG-VRP-100 [PUPO] on random dataset.

| Instance | Mean Time (s) | Mean Length | Mean Gap (%) | Min Gap (%) | Max Gap (%) | Std |
|---|---|---|---|---|---|---|
| U-50 | 0.178 | 10.811 | 9.093 | 2.543 | 22.455 | 0.031 |
| U-500 | 1.673 | 75.034 | 8.842 | 6.207 | 11.874 | 0.015 |
| U-5000 | 31.197 | 626.848 | 5.142 | 3.314 | 7.476 | 0.016 |
| C-50 | 0.189 | 9.359 | 8.047 | 1.617 | 23.159 | 0.038 |
| C-500 | 1.757 | 59.604 | 7.92 | 5.273 | 13.105 | 0.015 |
| C-5000 | 30.857 | 597.049 | 5.434 | 2.846 | 6.627 | 0.016 |
| E-50 | 0.19 | 9.647 | 9.148 | 1.991 | 23.308 | 0.034 |
| E-500 | 1.757 | 56.838 | 8.676 | 6.645 | 12.068 | 0.015 |
| E-5000 | 30.796 | 365.526 | 6.33 | 4.875 | 7.202 | 0.009 |
| I-50 | 0.189 | 10.262 | 8.673 | 1.927 | 23.732 | 0.036 |
| I-500 | 1.758 | 68.808 | 8.833 | 6.697 | 14.897 | 0.014 |
| I-5000 | 30.911 | 617.372 | 5.237 | 3.594 | 6.488 | 0.011 |

Table 24: Detailed results of INVIT-VRP-50 [Vanilla] on random dataset.

| Instance | Mean Time (s) | Mean Length | Mean Gap (%) | Min Gap (%) | Max Gap (%) | Std |
|---|---|---|---|---|---|---|
| U-50 | 0.836 | 10.286 | 4.251 | 0.544 | 8.944 | 0.016 |
| U-500 | 8.903 | 76.366 | 10.747 | 9.049 | 13.794 | 0.012 |
| U-5000 | 137.071 | 654.376 | 9.672 | 8.218 | 10.969 | 0.01 |
| C-50 | 0.74 | 8.978 | 4.5 | 0.718 | 15.507 | 0.024 |
| C-500 | 8.035 | 60.676 | 9.786 | 7.865 | 11.631 | 0.008 |
| C-5000 | 131.165 | 615.721 | 8.492 | 7.421 | 9.728 | 0.011 |
| E-50 | 0.769 | 9.199 | 4.555 | 0.296 | 11.336 | 0.018 |
| E-500 | 8.044 | 57.801 | 10.475 | 7.92 | 15.046 | 0.015 |
| E-5000 | 131.003 | 376.968 | 9.641 | 8.73 | 10.735 | 0.009 |
| I-50 | 0.743 | 9.775 | 4.408 | 0.427 | 12.283 | 0.019 |
| I-500 | 7.756 | 69.659 | 10.148 | 7.872 | 12.083 | 0.01 |
| I-5000 | 128.097 | 635.588 | 8.229 | 7.275 | 9.21 | 0.008 |

Table 25: Detailed results of INVIT-VRP-50 [PUPO] on random dataset.

| Instance | Mean Time (s) | Mean Length | Mean Gap (%) | Min Gap (%) | Max Gap (%) | Std |
|---|---|---|---|---|---|---|
| U-50 | 0.797 | 10.329 | 4.584 | 0.572 | 9.606 | 0.016 |
| U-500 | 7.909 | 75.738 | 9.828 | 7.848 | 12.912 | 0.011 |
| U-5000 | 128.458 | 639.748 | 7.256 | 6.153 | 9.235 | 0.012 |
| C-50 | 0.789 | 9.008 | 4.841 | 0.553 | 15.779 | 0.025 |
| C-500 | 7.947 | 60.189 | 8.926 | 7.654 | 11.551 | 0.009 |
| C-5000 | 127.466 | 608.262 | 7.072 | 6.281 | 7.985 | 0.007 |
| E-50 | 0.795 | 9.233 | 4.943 | 0.531 | 10.928 | 0.019 |
| E-500 | 8.012 | 57.44 | 9.819 | 7.607 | 14.179 | 0.014 |
| E-5000 | 132.614 | 370.181 | 7.705 | 6.471 | 9.107 | 0.012 |
| I-50 | 0.846 | 9.819 | 4.872 | 0.481 | 14.009 | 0.02 |
| I-500 | 8.1 | 69.219 | 9.49 | 7.007 | 12.117 | 0.012 |
| I-5000 | 130.251 | 627.491 | 6.873 | 6.444 | 7.233 | 0.003 |

Table 26: Detailed results of INVIT-VRP-100 [Vanilla] on random dataset.

| Instance | Mean Time (s) | Mean Length | Mean Gap (%) | Min Gap (%) | Max Gap (%) | Std |
|---|---|---|---|---|---|---|
| U-50 | 0.778 | 10.319 | 4.75 | 0.235 | 10.179 | 0.017 |
| U-500 | 7.906 | 75.658 | 9.672 | 8.112 | 11.936 | 0.009 |
| U-5000 | 130.972 | 643.717 | 7.896 | 6.829 | 9.479 | 0.01 |
| C-50 | 0.77 | 8.989 | 4.639 | 1.182 | 15.259 | 0.024 |
| C-500 | 7.903 | 60.362 | 9.223 | 7.537 | 11.981 | 0.009 |
| C-5000 | 126.556 | 611.67 | 7.655 | 7.229 | 7.876 | 0.003 |
| E-50 | 0.777 | 9.227 | 4.878 | 0.587 | 11.05 | 0.019 |
| E-500 | 7.915 | 57.486 | 9.82 | 7.829 | 12.352 | 0.01 |
| E-5000 | 127.763 | 372.908 | 8.472 | 7.441 | 9.643 | 0.01 |
| I-50 | 0.779 | 9.81 | 4.782 | 0.707 | 13.409 | 0.021 |
| I-500 | 7.941 | 69.15 | 9.336 | 7.609 | 11.371 | 0.009 |
| I-5000 | 127.558 | 630.358 | 7.353 | 6.908 | 7.836 | 0.004 |

Table 27: Detailed results of INVIT-VRP-100 [PUPO] on random dataset.

| Instance | Mean Time (s) | Mean Length | Mean Gap (%) | Min Gap (%) | Max Gap (%) | Std |
|---|---|---|---|---|---|---|
| U-50 | 0.776 | 10.365 | 5.055 | 0.369 | 9.866 | 0.017 |
| U-500 | 7.796 | 74.755 | 8.366 | 7.137 | 10.577 | 0.008 |
| U-5000 | 127.661 | 625.141 | 4.723 | 4.172 | 5.306 | 0.005 |
| C-50 | 0.775 | 9.017 | 4.973 | 0.651 | 17.35 | 0.026 |
| C-500 | 7.754 | 59.417 | 7.511 | 5.883 | 9.331 | 0.007 |
| C-5000 | 127.345 | 593.798 | 4.511 | 4.173 | 4.826 | 0.003 |
| E-50 | 0.783 | 9.27 | 5.365 | 0.804 | 12.195 | 0.021 |
| E-500 | 7.903 | 56.784 | 8.517 | 7.271 | 12.073 | 0.01 |
| E-5000 | 127.504 | 361.641 | 5.165 | 4.653 | 5.629 | 0.004 |
| I-50 | 0.793 | 9.856 | 5.283 | 0.473 | 13.671 | 0.021 |
| I-500 | 7.891 | 68.487 | 8.276 | 6.14 | 10.157 | 0.008 |
| I-5000 | 126.872 | 614.135 | 4.584 | 4.223 | 4.953 | 0.003 |

