# OpenReview forum: "Purity Law for Neural Routing Problem Solvers with Enhanced Generalizability"
_NeurIPS.cc/2025/Conference — NeurIPS 2025 poster_

### Official Review · Reviewer_Kzrh · 2025-06-30

**Clarity:** 3
**Significance:** 2
**Originality:** 3
**Rating:** 5
**Confidence:** 5

**Summary:**

The authors propose PUrity Policy Optimization (PUPO), a novel training framework that incorporates the Purity Law, a consistent structural pattern observed in routing problem solutions, into the policy optimization process. PUPO modifies the policy gradient to encourage the emergence of this structure during training, improving generalization across distributions and problem sizes without changing the model architecture. It is compatible with some existing neural solvers and enhances their performance without adding inference overhead.

**Questions:**

1. Is it possible to make a similar law for non-Euclidean problems, which are much more interesting and more important, because almost all real-life routing problems belong to this class?

2. The statistical properties of the proposed measure of optimal solutions (purity order) appear remarkably stable across the four instance distributions considered. However, since this remains a heuristic observation, could it be possible to characterize the class of distributions for which the purity law holds? Alternatively, could one learn or design a distribution of instances that maximally violates the purity law?

3. Given the apparent generality of the law, it is regrettable that it is leveraged only in a specific class of NCO algorithms (based on Reinforce). Another approach could be to incorporate purity orders as edge features in the instance input representation, making them accessible in an algorithm-agnostic way, whether REINFORCE-based or not, and even beyond RL? This could allow the algorithm to learn how best to exploit the proposed heuristic.

4. Related to the above, there is an alternative NCO approach known as the 'heavy-decoder/no-encoder' paradigm ([1, 2] and a few following works), which is known for its excellent generalization capabilities and relies on imitation learning. Is it possible to apply your method to such types of learning algorithms?
When talking about this paradigm, why did you not consider these works as baselines, especially given that they use the same experimental setup as your paper (training on TSP/VRP-100 and evaluating generalization)? Although they do not play with different instance distributions, comparisons on TSPLib and CVRPLib suggest they perform significantly better.

5. You trained the method on problems of size 50 and 100, but can you confirm that you trained 4 different policies for every distribution (U, C, E, I)? NCO methods claim to generalize to distribution shift, and it would therefore have been interesting to measure the deterioration of performance of, say, INViT-50 trained on U and tested on C, compared to INViT-50 trained on U and tested on U, especially to view how this deterioration evolves when adding PUPO. One would expect it to diminish with PUPO.


[1] Luo et al. Neural Combinatorial Optimization with Heavy Decoder: Toward Large Scale Generalization, NeurIPS 23

[2] Drakulic et al. BQ-NCO: Bisimulation Quotienting for Efficient Neural Combinatorial Optimization, NeurIPS 23

**Ethical Concerns:**

["NO or VERY MINOR ethics concerns only"]

**Final Justification:**

During the rebuttal, the authors addressed both of my concerns and demonstrated that this work is applicable beyond the cases described in the initial submission.

I clearly support its acceptance.

**Limitations:**

This work has two main limitations: it is applicable to Euclidean routing problems with the REINFORCE learning algorithm.

**Quality:**

3

**Strengths And Weaknesses:**

Strengths:
- The proposed measure is intuitive, but original in the area of NCO
- The statistical analysis is thoroughly conducted
- The paper is well written,  logically structured, and presents an interesting perspective.

Weaknesses:
- The proposed measure is limited to Euclidean problems and involves not only Euclidean distance but also the scalar product.
- The integration into the Reinforce algorithm is ad-hoc.

---

> ### Author Rebuttal · Authors · 2025-07-31
>
> Thank you very much for your time and effort in reviewing our work. We feel very glad to know you acknowledge our work is intuitive and original, find the statistical analysis is thoroughly conducted, and believe that our paper is well written, logically structured, and presents an interesting perspective.
>
> We will address each of your concerns in detail as follow.
>
> --Q1: Similar law for non-Euclidean problems?
>
>    Thank you for raising this important concern. In non-Euclidean problems, there is no longer direct connectivity between vertices, and the metric is given a priori by the edge cost matrix. Given $G=(V, E)$, we define purity order $K_p(e_{ij})$ in non-Euclidean routing problems as follow:
>
> $K _ p(e _ {ij}) = \frac{max _{m _ i \in M _ i} |\xi _ {im _ i}| + max _ {m _ j \in M _ j} |\xi _ {jm _ j}|}{2} $
> , where $M _ i $ consists of all $ m _i =  \arg\max _ {m, l(\xi _ {im}) \leq l(e _ {ij})/2}  l(\xi _ {im}) $,
>
> and $\xi_{im}$ is the shortest path from vertex $v_i$ to $v_m$.
>
> To understand it intuitively, we first need to find the shortest path starting from the two endpoints, which does not exceed half the length of the edge and has the largest length, and then define the purity order of the two end points as the average of the maximum number of nodes in the shortest path. We take an instance of the non-Euclidean TSP dataset of size 20 in [1] as an example and calculate the pure order according to the above definition. The purity order distribution is as follows:
>
>
>
> | Purity Order | Frequency | Purity Order | Frequency  |
> |--------------|-----------|--------------|------------|
> | 0            | 35        | 4            | 1          |
> | 1            | 11        | 5            | 0          |
> | 2            | 3         | 6            | 0          |
> | 3            | 0         | 7            | 0          |
>
>
> We can observe that the purity order still exhibits a negative exponential law on non-Euclidean TSP instances. In the future, we will calculate the purity order distribution of optimal solutions for more non-Euclidean TSP instances and add them to the appendix of this paper.
>
>
>
> --Q2: More theoretical discussion on purity law.
>
>   Thank you for this suggestion. Regarding your first point, we also find the stability of the purity law under different distributions interesting. We speculate that certain types of vertex distributions in instances may satisfy persistent homology in the field of topology, under measured by the "purity order on the optimal solution." However, we have not yet achieved a formal theoretical proof. In future work, we plan to explore more theoretical proofs of this relationship from the perspective of topological data analysis.
>
> Your second point is also interesting and inspiring. Taking the TSP as an example, for instances consisting of regular N-gon endpoints, the negative exponential law is not strictly satisfied. Because the purity order of the edges in the optimal solution is all 0. However, this can be viewed as a limiting case of the negative exponential law. Furthermore, we speculate that there should be special instances that theoretically violate the negative exponential law. In future work, we plan to theoretically construct such special instances or generate them through learning-based methods.
>
> Furthermore, we have rigorously proved an upper bound on the probability that an edge of purity order $k$ is located on the optimal solution under certain conditions. If you are interested, please refer to our response to Reviewer 8oG4.
>
>
>
> --Q3: Incorporating purity orders as instance input representation.
>
>   Thank you for this suggestion. On one hand, purity order can be incorporated as an edge feature into graph representation learning, such as Graph Convolutional Neural Networks architecture. On the other hand, purity order can be generalized from edges to certain vertex attributes to reflect the purity information of nodes in instances. In future work, we will explore more methods for incorporating purity information into representation learning.
>
>
> --Q4: Discussion on PUPO combined with heavy-decoder paradigm.
>
>   Thank you for this suggestion. The LEHD and BQ-NCO you mentioned are both excellent works, which use imitation learning for training. The loss function of imitation learning is as follows:
> $l = - \frac{1}{N} \sum_i (y_i \mathop{\log} p_i + (1-y_i) \mathop{\log} (1 - p_i) )$, where $y_i$ is optimal label.
>
> Following the PUPO in our paper, we propose the modified loss function as follow:
> $\hat{l} = - \frac{1}{N} \sum_i (y_i C(U_{i-1}, \tau_i) \mathop{\log} p_i + (1-y_i) C(U_{i-1}, \tau_i) \mathop{\log} (1 - p_i) ).$
>
> We then compare the two training methods using LEHD as an example, testing them on a few instances of TSPLib. The results are shown below.
>
> |          | PUPO   | Vanilla |        | PUPO   | Vanilla |         | PUPO   | Vanilla  |
> |----------|--------|---------|--------|--------|---------|---------|--------|----------|
> | a280     | 6.41   | 5.41    | d2103  | 15.33  | 19.11   | u1060   | 14.77  | 15.11    |
> | berlin52 | 0.04   | 0.02    | eil51  | 2.29   | 1.44    | u1432   | 11.53  | 10.93    |
> | bier127  | 6.84   | 6.54    | eil76  | 3.25   | 3.17    | u1817   | 13.71  | 13.87    |
> | ch130    | 2.83   | 2.77    | eil101 | 4.77   | 4.16    | u2152   | 20.14  | 25.58    |
> | ch150    | 11.68  | 8.99    | fl417  | 10.93  | 9.93    | u2319   | 6.38   | 7.27     |
> | d198     | 11.03  | 11.01   | fl1400 | 23.40  | 25.81   | vm1084  | 11.14  | 13.67    |
> | d493     | 9.16   | 9.76    | fl1577 | 23.65  | 28.23   | vm1748  | 17.55  | 18.11    |
> | d657     | 9.84   | 11.21   | gil262 | 2.73   | 2.75    | fl3795  | 44.19  | 48.12    |
> | d1291    | 15.26  | 19.32   | u159   | 1.79   | 1.66    | fnl4461 | 25.71  | 29.08    |
> | d1655    | 19.47  | 23.08   | u574   | 7.18   | 7.81    | pcb3038 | 17.64  | 21.74    |
>
>
> As can be seen, PUPO-training improves the model's generalization ability on larger scales, but may exhibit a trade-off on smaller instances. It's worth noting that due to limited time, neither method was fully trained. We believe that thorough training would further improve the performance of both models. A performance comparison of the two models after full training will be supplemented in the appendix.
>
>
>
>
> --Q5: More specific details on training.
>
>   Thank you for raising this concern. In this paper, all models are trained on a uniform distribution and then directly tested on the U, C, E, and I distributions. The test results in this paper reflect the performance changes caused by the distribution shift you mentioned. From Table 1 and Table 2 in this paper, PUPO-training improves the model's generalization ability to distribution shift and scale increase.
>
>
> Reference:
>
> [1] Matrix Encoding Networks for Neural Combinatorial Optimization, NeurIPS 2021

---

> > ### Comment · Reviewer_Kzrh · 2025-08-04
> >
> > Thank you for your response. My main concerns regarding applicability to non-Euclidean problems and other state-of-the-art NCO training paradigms have been addressed.
> >
> > It has been shown that this work is extendable to the non-Euclidean case and also improves NCO models under supervised and self-improvement training paradigms.
> >
> > I look forward to seeing detailed experiments on these extensions.
> >
> > With these improvements, I clearly support this paper.

---

> > > ### Author Response · Authors · 2025-08-05
> > >
> > > Thanks for your feedback and thoughtful consideration! We sincerely appreciate your valuable comments, prompt response, and recognition of our work.

---

### Official Review · Reviewer_VRAr · 2025-07-01

**Clarity:** 3
**Significance:** 3
**Originality:** 4
**Rating:** 5
**Confidence:** 4

**Summary:**

This paper proposes the purity law for neural routing solvers. Based on the purity law, the authors introduce purity policy optimization (PUPO) to enhance the generalization of the neural routing solvers. Experiments demonstrate that PUPO can be integrated into popular solvers and improve the generalization ability in routing problems, including TSP and CVRP.

**Questions:**

1.	Compared to Vanilla RL, PUPO uses purity weights in the loss function. I wonder whether the concept of purity law and purity weights can be applied to supervision-learning-based neural solvers.

**Ethical Concerns:**

["NO or VERY MINOR ethics concerns only"]

**Final Justification:**

The authors have addressed my concerns on the performance comparison with traditional solvers and deeper insights into the purity law.   I would like to remain the score.

**Quality:**

3

**Strengths And Weaknesses:**

This paper proposes a novel insight and an effective algorithm to solve combinatorial optimization. I would like to support the acceptance if the authors can address my concerns.

Strengths：

1.	The concept of purity law is impressive and interesting. The purity law is related to the generalizable structural patterns in routing problems, which is a novel insight.

2.	The proposed PUPO can be easily integrated with a wide range of existing neural routing solvers.

3.	The authors propose a solid theoretical analysis on the algorithm, introducing two key concepts purity availability and purity cost.

4.	The writing of the paper is good.

Weaknesses:

1.	The authors may want to include more baselines in the experiments. The performance of traditional solvers, such as LKH3 and Concorde, should be considered.

2.	In Tables 1 and 2, I find PUPO's performance worse than Vanilla. Since PUPO is trained on instances with scales 50 and 100, PUPO performs worse in the in-distribution instances. I encourage the authors to give a deeper explanation of the mechanism of implicit regularization.

---

> ### Author Rebuttal · Authors · 2025-07-31
>
> Thank you very much for your time and effort in reviewing our work. We feel very glad to know you acknowledge our purity is novel and impressive, find the theoretical analysis is solid, and believe that our paper is well written.
>
> We will address each of your concerns in detail as follow.
>
> --Q1: Including concorde in the experiments.
>
> Thank you for raising this concern. The test datasets used in this paper is from [1]. In the datasets with multi-distribution, TSP-100 is solved by Gurobi, an exact solver, so the solutions are guaranteed to be truly optimal. TSP-1000, TSP-5000 and TSP-10000 are solved by a SOTA heuristic algorithm LKH3, which can output near-optimal solutions.
> To settle down your concerns, we use concorde to resolve three large datasets. We report the gap between LKH3 and concorde on TSP-1000 as follow, where negative gap means concorde performs better than LKH3.
>
> |           | Uniform      | Clustered    | Explosion   | Implosion    |
> |-----------|--------------|--------------|-------------|--------------|
> | Cost      | 23.164  | 14.095  | 16.052 | 20.169   |
> | Gap(\%) | -0.248 | -0.016 | -0.073 | -0.102  |
>
>
> It can be observed that LKH3 can reach the optima very nearly. However, due to unacceptable time consumption (one instance need more than 20 hours), concorde fails on the TSP-5000 and TSP-10000. In the future, we will make the concorde-solved optimal values and solutions of each of 200 instances with four distributions in TSP-1000 all open source.
>
>
> --Q2: Deeper explanation of the mechanism of implicit regularization.
>
> Thank you for this concern. The slight performance decline on small-scale problems derives from the underlying mechanism of implicit regularization of PUPO. Shown in Table 5, we can observe that the PUPO training reduces the sum of the parameter Frobenius norms. It results in an expected but reasonable trade-off on small, identically distributed datasets.
>
> When the scale and distribution of training data is limited and completely same with the test data, PUPO-trained model may underfit the specific patterns and noise of the training set. This is because regularization prevents the model from excessively memorizing "details" of the training data, which might be important for decision-making in a small, same-distributed setting. Vanilla-trained models are more likely to achieve near-optimal fit to the training set in this "ideal" small, same-distributed setting.
>
> However, during vanilla training, tour length serves as the sole reward signal. Lacking guidance from generalizable structure, the model often overfits to these specific patterns in the small training set, resulting in weak generalization. PUPO introduces purity information into the model training process, allowing the model to focus on generalizable patterns even in the small training set, significantly improving cross-distribution and larger-scale generalizability, which is potentially more challenging and relevant in the real world.
>
> On multiple large-scale datasets (1000, 5000, and 10000), our algorithm achieves significant and consistent enhancements compared to vanilla training. In particular, as shown in Table 2, the PUPO-trained ELG-50 achieves a 68.58% improvement on the E-5000 of CVRP. On the distribution-shifted test sets (C, E, I, TSPLib and CVRPLib), the PUPO-trained model also achieves significant cross-distribution generalization, demonstrating the effectiveness of the implicit regularization introduced by PUPO in addressing data distribution-shifted generalization.
>
> In future work, to balance the model's performance on small datasets, we will introduce a tuning parameter $\alpha \geq 0$ and generalize the Purity Weightings to $\hat{W}(U_t, \tau_{t+1}) = W^{\alpha}(U_t, \tau_{t+1})$ to adjust the implicit regularization degree. When $\alpha$ reaches 0, PUPO degenerates to a vanilla policy gradient.
>
>
>
>
> --Q3: More discussion on supervision-learning-based neural solvers.
> Thank you for this suggestion. For the supervision-learning-based neural solvers, their loss function of imitation learning is as follows:
> $l = - \frac{1}{N} \sum_i (y_i \mathop{\log} p_i + (1-y_i) \mathop{\log} (1 - p_i) )$, where $y_i$ is optimal label.
>
> Following the PUPO in our paper, we propose the modified loss function as follow:
> $\hat{l} = - \frac{1}{N} \sum_i (y_i C(U_{i-1}, \tau_i) \mathop{\log} p_i + (1-y_i) C(U_{i-1}, \tau_i) \mathop{\log} (1 - p_i) ).$
>
> We then compare the two training methods using LEHD [2] as an example, testing them on a few instances of TSPLib. The results are shown below.
>
> |          | PUPO   | Vanilla |        | PUPO   | Vanilla |         | PUPO   | Vanilla  |
> |----------|--------|---------|--------|--------|---------|---------|--------|----------|
> | a280     | 6.41   | 5.41    | d2103  | 15.33  | 19.11   | u1060   | 14.77  | 15.11    |
> | berlin52 | 0.04   | 0.02    | eil51  | 2.29   | 1.44    | u1432   | 11.53  | 10.93    |
> | bier127  | 6.84   | 6.54    | eil76  | 3.25   | 3.17    | u1817   | 13.71  | 13.87    |
> | ch130    | 2.83   | 2.77    | eil101 | 4.77   | 4.16    | u2152   | 20.14  | 25.58    |
> | ch150    | 11.68  | 8.99    | fl417  | 10.93  | 9.93    | u2319   | 6.38   | 7.27     |
> | d198     | 11.03  | 11.01   | fl1400 | 23.40  | 25.81   | vm1084  | 11.14  | 13.67    |
> | d493     | 9.16   | 9.76    | fl1577 | 23.65  | 28.23   | vm1748  | 17.55  | 18.11    |
> | d657     | 9.84   | 11.21   | gil262 | 2.73   | 2.75    | fl3795  | 44.19  | 48.12    |
> | d1291    | 15.26  | 19.32   | u159   | 1.79   | 1.66    | fnl4461 | 25.71  | 29.08    |
> | d1655    | 19.47  | 23.08   | u574   | 7.18   | 7.81    | pcb3038 | 17.64  | 21.74    |
>
> As can be seen, PUPO-training improves the model's generalization ability on larger scales, but may exhibit a trade-off on smaller instances. It's worth noting that due to limited time, neither method was fully trained. We believe that thorough training would further improve the performance of both models. A performance comparison of the two models after full training will be supplemented in the appendix.
>
> Reference:
>
> [1] INViT: A Generalizable Routing Problem Solver with Invariant Nested View Transformer, ICML 2024
>
> [2] Neural Combinatorial Optimization with Heavy Decoder: Toward Large Scale Generalization, NeurIPS 23

---

> > ### Comment · Reviewer_VRAr · 2025-08-02
> >
> > Thanks for the rebuttal from the authors! The rebuttal has addressed my concerns. I think this paper is a technically solid paper with interesting observations. Despite the concerns from the other reviewers, I support the paper.

---

> > > ### Author Response · Authors · 2025-08-02
> > >
> > > Thanks for your feedback and thoughtful consideration! We sincerely appreciate your valuable comments, prompt response, and recognition of our work.

---

### Official Review · Reviewer_iRH9 · 2025-07-02

**Clarity:** 3
**Significance:** 2
**Originality:** 3
**Rating:** 4
**Confidence:** 4

**Summary:**

This paper introduces Purity Law, a principle revealing that edge prevalence in optimal solutions of routing problems grows exponentially with the sparsity of surrounding vertices. The authors propose Purity Policy Optimization (PUPO) that aligns NCO solvers with this law to enhance the generalization ability. On TSP and CVRP tasks, they show that PUPO integrates seamlessly with popular neural solvers (e.g., ELG, INViT) to improve performance on large-scale and heterogeneous problems without increasing inference costs.

**Questions:**

1. How does the proven supermodularity property guide the algorithm design? Could you explain this in more detail? Furthermore, can this supermodularity property lead to theoretical approximation ratios?
2. Can the purity cost be directly incorporated into the reward? Although the current purity cost method considers multi-step accumulation, the lack of a baseline or critic may result in high variance. Would integrating purity cost as part of the reward yield better results?

**Ethical Concerns:**

["NO or VERY MINOR ethics concerns only"]

**Final Justification:**

Some of my concerns have been addressed. I decide to weakly accept it.

**Limitations:**

Yes

**Paper Formatting Concerns:**

No.

**Quality:**

2

**Strengths And Weaknesses:**

Strengths
1. The motivation of this paper is clear. The author first observed a common purity pattern in large-scale TSP instances and was inspired by this to propose a purity cost to guide policy training, encouraging the policy to favor solutions with a smaller overall purity order, achieving improvements in generalization performance in experiments.
2. The proposed algorithm is novel and effective. The key technical contributions lie in uncovering a universal structural pattern and leveraging it to guide policy learning, demonstrating significant generalization gains in empirical evaluations. It make sense to me.

Weaknesses
1. Many experimental data in the paper are inconsistent with those in related works. For example, in the TSP100 data,  INViT reports a 0.95% gap and ELG reports a 0.22% gap, which differ significantly from the 5.86% and 2.61% shown in Table 1 of this paper. After comparing most of the experimental results in Table 1 and Table 2, I found substantial discrepancies with existing results.
2. Based on the current experimental results, the proposed method appears to impair performance on small-scale problems.
3. Although the paper proposes some theoretical aspects, they provide limited guidance for algorithm design. In particular, it lacks theoretical analysis of relevant performance metrics, such as the algorithm’s approximation ratio, sample complexity analysis related to reinforcement learning, or classical generalization error analysis, which limits the theoretical contributions of the paper. I would like to see more discussion related to algorithm design.

---

> ### Author Rebuttal · Authors · 2025-07-31
>
> Thank you very much for your time and effort in reviewing our work. We feel glad to know you acknowledge our work is novel and effective, and believe our presenting and motivation are clear.
>
> We will address each of your concerns in detail as follow.
>
> --Q1: Inconsistence between experimental data in this paper with those in related works?
>
> Thanks for your thorough review. In our experiments, to ensure fairness, we conduct each training process for the model under the same hardware conditions and parameters (except for the learning rate). In the original INViT paper, all numerical experiments are performed on an NVIDIA RTX 4090. While in the original ELG paper, models are trained on NVIDIA RTX 6000 Ada with 48GB memory, tested on one NVIDIA RTX 4090. Whereas our experiments are implemented on an NVIDIA RTX 3090. Additionally, the selection of the starting point for the tour is random. These factors may account for the slight differences observed when compared to the results in related works. If the paper is accepted, we will make all the code and detailed training logs open source.
>
>
>
> --Q2: Performance decline on small-scale problems.
>
> Thanks for your thorough review. The slight performance decline on small-scale problems derives from the underlying mechanism of implicit regularization of PUPO. Shown in Table 5, we can observe that the PUPO training reduces the sum of the parameter Frobenius norms. It results in an expected but reasonable trade-off on small, identically distributed datasets.
>
> When the scale and distribution of training data is limited and completely same with the test data, PUPO-trained model may underfit the specific patterns and noise of the training set. This is because regularization prevents the model from excessively memorizing "details" of the training data, which might be important for decision-making in a small, same-distributed setting. Vanilla-trained models are more likely to achieve near-optimal fit to the training set in this "ideal" small, same-distributed setting.
>
> However, during vanilla training, tour length serves as the sole reward signal. Lacking guidance from generalizable structure, the model often overfits to these specific patterns in the small training set, resulting in weak generalization. PUPO introduces purity information into the model training process, allowing the model to focus on generalizable patterns even in the small training set, significantly improving cross-distribution and larger-scale generalizability, which is potentially more challenging and relevant in the real world.
>
> On multiple large-scale datasets (1000, 5000, and 10000), our algorithm achieves significant and consistent enhancements compared to vanilla training. In particular, as shown in Table 2, the PUPO-trained ELG-50 achieves a 68.58% improvement on the E-5000 of CVRP. On the distribution-shifted test sets (C, E, I, TSPLib and CVRPLib), the PUPO-trained model also achieves significant cross-distribution generalization, demonstrating the effectiveness of the implicit regularization introduced by PUPO in addressing data distribution-shifted generalization.
>
> In future work, to balance the model's performance on small datasets, we will introduce a tuning parameter $\alpha \geq 0$ and generalize the Purity Weightings to $\hat{W}(U_t, \tau_{t+1}) = W^{\alpha}(U_t, \tau_{t+1})$ to adjust the implicit regularization degree. When $\alpha$ reaches 0, PUPO degenerates to a vanilla policy gradient.
>
>
> --Q3: More discussion related to algorithm design.
>
> Thank you for this suggestion. We supplemented the theoretical analysis on optimization error of PUPO. First, we denote
> $\nabla V^{p_{\theta}} = E_{p_{\theta}(\tau | \mathcal{X})} \Biggl[   \sum \limits_{t=2}^{N} \left( L(\tau) - b(X) \right) W(U_{t-1}, \tau_{t}) \nabla \log p_{\theta}(\tau_{t} | \tau_{1:t-1}, X)  \Biggr]$,
>
> and $\widehat{\nabla V^{p_{\theta}}} = \sum \limits_{t=2}^{N} \left( L(\tau) - b(X) \right) W(U_{t-1}, \tau_{t}) \nabla \log p_{\theta}(\tau_{t} | \tau_{1:t-1}, X) $.
>
> The update rule of gradient ascent algorithm is $\theta_{t+1} = \theta_t + \eta_t \widehat{\nabla V^{p_{\theta}}}$.
>
>
> Let us say a function $f$ is $\beta$-smooth if $||\nabla f(\theta)-\nabla f(\theta')|| \leq \beta||\theta-\theta'||$, where the norm $||\cdot||$ is the Euclidean norm.
> Assume that for all $\theta$, $V^{p_{\theta}}$ is $\beta$-smooth and bounded $V_{\star}$. Suppose the variance is bounded as follows:
> $E [ ||\widehat{\nabla V^{p_{\theta}}}-\nabla V^{p_{\theta}}||^2 ] \leq \sigma^2$.
> For $t \leq \beta (V^*-V^{(0)})/\sigma^2$, suppose we use a constant stepsize of $\eta_t = 1/\beta$, and thereafter, we use $\eta_t = \sqrt{2/(\beta T)}$. For all $T$, we can obtain:
>
> $\min\limits_{t \leq T} E[||\nabla V^{(t)}||^2] \leq \frac{2 \beta (V^*-V^{(0)})}{T} + \sqrt{\frac{2\sigma^2}{T}}$
>
> The proof method is similar to [1]. And this optimization error theorem confirms that our proposed algorithm can converge to stationary points.
>
> However, due to the black-box nature of neural networks, proving the approximation ratio and generalization error of learning methods is extremely difficult. The gap evaluation metric in this paper can experimentally reflect the function of the approximation ratio, showing the gap ratio between model performance and the optimal solution. In future work, we hope to provide more comprehensive algorithm design analysis from a theoretical perspective. The above optimization error analysis will be added to the appendix of this paper.
>
> --Q4: Guidance role of supermodularity property to algorithm design.
>
> Thank you for raising this concern. In the late stages of the neural constructive solver's decision-making process, the set of unvisited nodes is quite narrow. Because the agent's early stages occupy underlying better future node selection options, it is likely to choose poor ones that are detrimental to generalization. However, in vanilla training, tour length is the only reward signal, which fails to identify this impact of late-stage decisions. These late-stage decisions, made within a narrow feasible set, are crucial for generalization. The increasing marginal returns of the supermodular function, Purity Availability, effectively capture this tail effect.
>
> Inspired by this motivation, we constructed a Purity Availability metric with supermodular properties to measure the purity potential of the unvisited vertex set. On one hand, since Purity Availability represents the average minimum available purity order, higher values indicate that the state will not be able to generate highly pure structures in the future. For example, in the Figure 5 in appendix, the Purity Availability of strategies (a) and (b) at the current state are 1 and 0, respectively. It means that policy (b) possess ability to construct purer structures in the future. On the other hand, constrained by the necessity to select nodes only from the unvisited vertex set, later decision stages often lead to states that are unfavorable for generalization and difficult to distinguish under tour length metrics.
>
> Due to supermodularity, the marginal benefit of Purity Availability, $\phi (U_t) - \phi (U_{t-1})$, is greater at later stages, making the purity measure in the later decision phases more significant. Inspired by supermodularity, we designed the purity cost based on the marginal benefit of state and the purity order of the actions. Furthermore, we designed purity weights based on the concept of future discounting to comprehensively assess the future purity potential of intermediate states during policy transitions.
>
>
> --Q5: Incorporating purity cost directly into the reward.
>
> Thank you for this suggestion. We use the sum of the pure orders of the edges in the tour as a regularization term and design the following modified loss function: $\hat{L}(\tau) = L(\tau) + \lambda \sum_{(\tau_i, \tau_j) \in \tau}K_p(\tau_i, \tau_j).$
> Based on INViT-50-TSP, while maintaining the same experimental settings as in our paper, we trained the model with $\lambda$ set to 1, 0.1, and 0.01. A larger $\lambda$ indicates a higher importance of purity information in the loss function. The test results are as follows.
>
> |         | $\lambda$ = 0.01   | $\lambda$ = 0.1    | $\lambda$ = 1      |         | $\lambda$ = 0.01   |$\lambda$ =  0.1    | $\lambda$ = 1       |
> |---------|--------|--------|--------|---------|--------|--------|---------|
> | U-100   | 2.37   | 3.26   | 5.01   | E-100   | 2.37   | 3.18   | 4.69    |
> | U-1000  | 7.77   | 10.26  | 14.38  | E-1000  | 9.92   | 12.31  | 16.83   |
> | U-5000  | 9.88   | 13.03  | 18.69  | E-5000  | 12.51  | 15.14  | 20.99   |
> | U-10000 | 8.31   | 11.62  | 17.12  | E-10000 | 11.31  | 14.74  | 19.59   |
> | C-100   | 3.04   | 3.51   | 4.75   | I-100   | 2.59   | 3.30   | 4.72    |
> | C-1000  | 8.89   | 11.01  | 14.24  | I-1000  | 8.14   | 10.44  | 14.53   |
> | C-5000  | 10.56  | 14.16  | 17.98  | I-5000  | 10.16  | 13.41  | 18.82   |
> | C-10000 | 11.08  | 14.04  | 18.73  | I-10000 | 8.85   | 12.67  | 18.47   |
>
>
> We can see that the test results for all three models are worse than those obtained with PUPO training. The degree of deterioration increases with increasing $\lambda$. Explicitly incorporating purity information into the loss function can lead to a mix of purity information and length information, which can lead to situations where the degree of purity improves but length deteriorates, making it impossible to distinguish well between them. This prevents the model from effectively perceiving purity information during training.
>
>
>
> Reference:
>
> [1] Reinforcement Learning: Theory and Algorithms. Alekh Agarwal, Nan Jiang, Sham M. Kakade, Wen Sun

---

> > ### Comment · Reviewer_iRH9 · 2025-08-04
> >
> > Thanks for your detailed response. Some of my concerns have been addressed. I decide to weakly accept it with the promised changes of the paper.

---

> > > ### Author Response · Authors · 2025-08-05
> > >
> > > Thanks for your feedback and thoughtful consideration! We sincerely appreciate your valuable comments, prompt response, and recognition of our work.

---

### Official Review · Reviewer_8oG4 · 2025-07-03

**Clarity:** 2
**Significance:** 2
**Originality:** 2
**Rating:** 3
**Confidence:** 4

**Summary:**

The paper proposes an empirical observation for routing problems in a metric space, introducing the concept of purity, which roughly measures the number of vertices in the neighborhood of an edge within the optimal solution of a minimum-cost routing problem (e.g., TSP or CVRP). Observing empirically that optimal solutions tend to have low purity on synthetic instances, the authors propose incorporating purity as a reward signal for reinforcement learning. Experiments on both synthetic and standard datasets for TSP and CVRP suggest that adding purity to the reward improves over vanilla REINFORCE.

**Questions:**

Do the authors have any sense of whether the purity phenomenon could be formally proven, at least for random Euclidean TSP? It seems plausible that one could rigorously show that cost-optimal solutions generally avoid long edges spanning densely populated regions.

**Ethical Concerns:**

["NO or VERY MINOR ethics concerns only"]

**Final Justification:**

The paper is substantially improved with theoretical results that prove its primary claim.  That a provable property of euclidean, random, routing problems can be demonstrated to offer consistent improvements over a vanilla baseline is conceptually clear and elegant.  The comparisons are now thorough.  As such, I've decided to increase my score.

**Limitations:**

Yes

**Quality:**

2

**Strengths And Weaknesses:**

Strengths: Identifying reward signals that improve reinforcement learning in routing problems is a valuable contribution. The notion of purity is simple yet elegant, and the underlying intuition — that a cost-minimizing tour is unlikely to use edges spanning densely populated regions — is sound. That this observation can be transformed into a useful reward signal helps shed light on what kinds of dense, structured rewards are more amenable to policy gradient methods. This kind of insight may well generalize beyond routing to other combinatorial settings.

Weakness: The main weakness is that REINFORCE itself is a relatively weak baseline. While I appreciate the simplicity of evaluating the method with REINFORCE on TSP and with a random graph baseline, this evaluation alone may not fully demonstrate the potential of the proposed reward shaping. I encourage the authors to also consider how their method compares against classical solvers such as concorde.  It's ok to lose, but it's informative to know by how much.

---

> ### Author Rebuttal · Authors · 2025-07-31
>
> Thank you very much for your time and effort in reviewing our work. We feel glad to know you acknowledge our work is a valuable contribution, find notion of purity is a sound underlying intuition, simple yet elegant, and believe this work would be insightful for the community.
>
> We will address each of your concerns in detail as follow.
>
> --Q1: Integration with REINFORCE.
>
> Thank you for raising this concern. First, the proposed PUPO can be combined with any policy-based method. Due to the fundamental role of REINFORCE in NCO, we first combined PUPO with it and found that after PUPO-training, the model's cross-distribution and large-scale generalizabilities were significantly improved. Furthermore, we conducted experiments combining PUPO with imitation learning, and the results were further improved. (Please see our response to Reviewer Kzrh.)
>
> In the field of neural routing problems, REINFORCE is a principal training method for neural constructive approaches, due to its ease of instance generation for the routing problems and its efficient sample utilization for reward computation. It is still adopted by the latest and most advanced works, such as [1-7].
>
> Traditional reinforcement learning environments require CPU-based trajectory data generation, which is very time-consuming and slow. Therefore, advanced algorithms such as PPO are needed to improve sample utilization efficiency. However, in the neural routing problems, instance generation is very simple and easy to parallelize on GPUs, and reward computation can efficiently utilize samples. At the same time, [8] demonstrates that the complex hyper parameter tuning of PPO may slightly hurt performance. Also [8] shows that REINFORCE-trained model performs even better than PPO-trained on mDPP. Therefore, REINFORCE is a vital baseline for neural routing problem solvers training.
>
> --Q2: Comparison against concorde.
>
> Thank you for this suggestion. The test datasets used in this paper is from [6]. In the datasets with multi-distribution, TSP-100 is solved by Gurobi, an exact solver, so the solutions are guaranteed to be truly optimal. TSP-1000, TSP-5000 and TSP-10000 are solved by a SOTA heuristic algorithm LKH3, which can output near-optimal solutions.
>
> To settle down your concerns, we use concorde to resolve three large datasets. We report the gap between LKH3 and concorde on TSP-1000 as follow, where negative gap means concorde performs better than LKH3.
>
> |           | Uniform      | Clustered    | Explosion   | Implosion    |
> |-----------|--------------|--------------|-------------|--------------|
> | Cost      | 23.164  | 14.095  | 16.052 | 20.169   |
> | Gap(\%) | -0.248 | -0.016 | -0.073 | -0.102  |
>
> It can be observed that LKH3 can reach the optima very nearly. However, due to unacceptable time consumption (one instance need more than 20 hours), concorde fails on the TSP-5000 and TSP-10000.
>
> In the future, we will make the concorde-solved optimal values and solutions of each of 200 instances with four distributions in TSP-1000 all open source.
>
>
> --Q3: Formal proof for purity phenomenon.
>
> Thank you for raising this important concern. For the random Euclidean TSP, we theoretically prove that under certain conditions, the probability of an edge with purity order $k$ being at the optimal solution is bounded by $\frac{k!!}{4^{\frac{k}{2}-1}(k+1)!!}$. Notably, this upper bound decreases monotonically with increasing $k$. This means that edges of higher purity order have a lower probability of forming the optimal solution, consistent with our findings. Furthermore, the negative exponential form of this upper bound indirectly confirms the purity law we propose. If the paper is accepted, the theorem and detailed analysis and proof will be added to the paper.
>
>
> THEOREM: For edge $e_{xy}$, assume that $K_p(e_{xy})=k$ is an positive even number, $z_i \in N_c(e_{xy})$ evenly distributed on both sides of $e_{xy}$, and the probability of each node being selected follows a uniform distribution.
> Then, the probability of $e_{xy}$ being at the optimal solution holds the following inequality:
>
> $P[e_{xy} \in \tau_{OPT}] \leq \frac{k!!}{4^{\frac{k}{2}-1}(k+1)!!}.$
>
>  Let $\omega_{xy}$ denote the Hamilton Path formed by edge $e_{xy}$ and vertices $z_i \in N_c(e_{xy})$, $\tau_{\omega_{xy}}$ denote the Hamilton Circle including $\omega_{xy}$. Then, the following inequality holds:
>
>  $P[e_{xy} \in \tau_{OPT}] \leq P[$ edges in $\tau_{\omega_{xy}}$ do not cross $] \leq P[$edges in $\omega_{xy}$ do not cross $]$.
>
> In the following, We will analyze the probability that no edges in $\omega_{xy}$ cross and obtain an upper bound. We prove the theorem by induction.
>
> When $K_p(e_{xy})=k=2$, it can be proven by enumeration method.
>
> Now, we first assume that when $K_p(e_{xy})=k$, the following equation holds:
>
> P[ edges in $\omega_{xy}$ do not cross $] = \frac{k!!}{4^{k/2-1} (k+1)!!}$.
>
>
> It means that among all $(k+1)!$ combinations of $\omega_{xy}$, there are  $\frac{(k!!)^2}{4^{{k/2}-1}}$ situations where the edges of $\omega_{xy}$ do not cross.
>
>
> When $K_p(e_{xy})=k+2$, first, there are totally $(k+3)!$ combinations of $\omega_{xy}$. Next, we consider the non-intersection structure based on case $k$.
> For every $\frac{(k!!)^2}{4^{{k/2}-1}}$ situations where the edges of $\omega_{xy}$ do not cross, consider one side of $e_{xy}$ first. If adding a point still leaves no intersection, there are $k/2+1$ possible addition positions. Considering both sides symmetrically, there are $(k/2+1)^2=\frac{(k+2)^2}{4}$ non-intersection cases. Therefore,
>
> P[ edges in $\omega_{xy}$ do not cross $] = \frac{\frac{(k!!)^2}{4^{{k/2}-1}} \cdot  \frac{(k+2)^2}{4}}{(k+3)!} = \frac{(k+2)!!}{4^{(k+2)/2-1} ((k+2)+1)!!}$.
>
> Hence, we have finished the proof of
> $P[e_{xy} \in \tau_{OPT}] \leq \frac{k!!}{4^{\frac{k}{2}-1}(k+1)!!}$.
>
> Reference:
>
> [1] CaDA: Cross-Problem Routing Solver with Constraint-Aware Dual-Attention, ICML 2025
>
> [2] SHIELD: Multi-task Multi-distribution Vehicle Routing Solver with Sparsity and Hierarchy, ICML 2025
>
> [3] Rethinking Light Decoder-based Solvers for Vehicle Routing Problems, ICLR 2025
>
> [4] PolyNet: Learning Diverse Solution Strategies for Neural Combinatorial Optimization, ICLR 2025
>
> [5] UDC: A Unified Neural Divide-and-Conquer Framework for Large-Scale Combinatorial Optimization Problems, NeurIPS 2024
>
> [6] INViT: A Generalizable Routing Problem Solver with Invariant Nested View Transformer, ICML 2024
>
> [7] Towards Generalizable Neural Solvers for Vehicle Routing Problems via Ensemble with Transferrable Local Policy, IJCAI 2024
>
> [8] RL4CO: an Extensive Reinforcement Learning for Combinatorial Optimization Benchmark

---

> > ### Comment · Reviewer_8oG4 · 2025-08-06
> >
> > The new proof of the purity law for the euclidean tour is a nice addition to the paper.  Thank you for your response to Q1, clarifying reinforce vs. PPO for neural routing problems.  I find the contribution substantially improved in light of the rebuttal, and have decided to increase my score.  I strongly encourage the authors to include the new proof in their final submission as it greatly strengthens their arguments.

---

> > > ### Author Response · Authors · 2025-08-08
> > >
> > > Thanks for your feedback and thoughtful consideration! We sincerely appreciate your valuable comments and recognition of our work.

---

> ### Author Response · Authors · 2025-08-05
>
> Dear Reviewer 8oG4,
>
> We sincerely appreciate your thoughtful comments and suggestions again.
>
> We would like to kindly follow up to check whether our response has addressed your concerns. If there are any remaining questions or suggestions, we would greatly appreciate the opportunity to clarify further.
>
> If our responses have satisfactorily addressed your questions, we would be grateful if you would consider revisiting your rating.
>
> Thank you once again for your time and constructive feedback.
>
> Best regards,
>
> The Authors of Submission 19649

---

### Decision · Program_Chairs · 2025-09-17

**Decision:**

Accept (poster)

**Comment:**

This work proposes and investigates a purity law for the routing problem, which is an interesting and important observation that "the proportion of different edges in the optimal solution follows a negative exponential law based on their purity orders across various instances," where the purity order is defined as a measure of vertex density around edges. During the rebuttal, a formal analysis is provided to further support this observation for Euclidean routing problems with random distribution. Based on the purity law, this work further proposes the Purity Policy Optimization (PUPO) training paradigm to improve the generalization ability for neural solvers. Experimental results show that PUPO can be integrated with different neural routing solvers to improve their generalization performance.

The reviewers find this paper well-written with clear motivation, the introduced purity law intuitive, elegant, and impressive, the proposed PUPO method novel, effective, and can be easily integrated into different neural routing solvers, and the experimental studies thorough with strong results. Some concerns have been raised on the REINFORCE training method, theoretical analysis, extension to non-Euclidean problems, extension to the "heavy-decoder/encoder" paradigm, experimental settings, and results. After the rebuttal, most of these concerns have been properly addressed. Finally, two reviewers vote to clearly accept this work, one reviewer leans toward weak acceptance, and one reviewer recommends a weak rejection. In the final justification, the most negative reviewer believes this paper is "substantially improved with theoretical results that prove its primary claim", the provable property is "conceptually clear and elegant", and the comparisons are "now thorough".

I read the paper myself and fully agree with the reviewers that the introduced purity law is novel and impressive, and the proposed PUPO training method is effective with strong generalization performance. I appreciate the formal theoretical analysis provided in the rebuttal to support the observation of purity law, and believe this analysis has substantially improved the quality and impact of this work. Similar to Reviewer Kzrh, I also look forward to seeing the extended results on non-Euclidean problems and the family of BQ-NCO/LEHD methods on much larger scale instances. Therefore, I recommend a clear acceptance of this work.

Please make sure all the discussion and revisions are carefully incorporated into the final paper, and make all the code and detailed training logs open source as promised.